

# Air quality in the Kathmandu Valley: WRF and WRF-Chem simulations of meteorology and black carbon concentrations

Andrea Mues[1], Axel Lauer[2], Aurelia Lupascu[1], Maheswar Rupakheti[1], Friderike Kuik[1], and Mark G. Lawrence[1]

[1]Institute for Advanced Sustainability Studies (IASS), Potsdam, 14467, Germany
[2]Deutsches Zentrum für Luft- und Raumfahrt (DLR), Institut für Physik der Atmosphäre, Oberpfaffenhofen, Germany

*Correspondence to:* A. Mues (Andrea.Mues@iass-potsdam.de) and A.Lauer (Axel.Lauer@dlr.de)

**Abstract.** An evaluation of the meteorology simulated using the Weather Research and Forecast (WRF) model for the region South Asia and Nepal with a focus on the Kathmandu Valley is presented. A particular focus of the model evaluation is placed on meteorological parameters that are highly relevant to air quality such as wind speed and direction, boundary layer height and precipitation. The same model setup is then used for simulations with WRF including chemistry and aerosols (WRF-Chem).

A WRF-Chem simulation has been performed using the state-of-the-art emission database EDGAR HTAP v2.2, along with a sensitivity simulation using observation-based black carbon emission fluxes for the Kathmandu Valley. The WRF-Chem simulations are analyzed in comparison to black carbon measurements in the valley and to each other.

The evaluation of the WRF simulation with a horizontal resolution of 3 x 3 $km^2$, shows that the model is often able to capture important meteorological parameters inside the Kathmandu Valley and the results for most meteorological parameters are well within the range of biases found in other WRF studies especially in mountain areas. But the evaluation results also

clearly highlight the difficulties of capturing meteorological parameters in such complex terrain and reproducing subgrid-scale processes with a horizontal resolution of 3 x 3 $km^2$. The measured black carbon concentrations are typically systematically and strongly underestimated by WRF-Chem. A sensitivity study with improved emissions in the Kathmandu Valley shows significantly reduced biases but also underlines several limitations of such corrections. Further improvements of the model and of the emission data are needed before being able to use the model to robustly assess air pollution mitigation scenarios in the

Kathmandu region.

## 1 Introduction

Severe air pollution has become an increasingly important problem in Nepal, in particular in the highly populated area of the Kathmandu Valley where about 12 % of the entire population of Nepal lives. Despite the air quality problems related to the

rapid population growth and the associated additional anthropogenic emissions in the valley, long-term measurements of air pollutants in the Kathmandu Valley were not made until recently. In collaboration with scientists from nearly 20 different research institutions in different countries, an atmospheric characterization campaign (SusKat-ABC – A Sustainable Atmosphere for the Kathmandu Valley, endorsed by the Atmospheric Brown Cloud (ABC) Programme of the United Nations Environment





Programme (UNEP)) measuring meteorological parameters and air pollutants in Nepal with a focus on the Kathmandu Valley was conducted from December 2012 through June 2013 (Rupakheti et al., 2017). The measurement results obtained during SusKat-ABC highlight the severe air pollution and the need for a better understanding of the emissions as well as of the meteorological and chemical processes resulting in such high pollution levels in the valley. Modeling studies using regional
atmospheric chemistry models with sufficiently high spatial resolution (e.g., 3 x 3 $km^2$ over the valley) to start resolving key features of the very complex topography in this region can support the analysis and interpretation of the measurement results. Here, first simulations covering the January to June 2013 period during the SusKat-ABC campaign with the Weather Research and Forecasting Model (WRF) (Skamarock et al., 2008) and a WRF version including chemistry and aerosols (WRF-Chem) (Fast et al., 2006; Grell et al., 2005) are performed in the framework of the projects SusKat and BERLiKUM (An assessment
of the impact of black carbon on air quality and climate in the Kathmandu Valley and surroundings – a model study). Previous model studies on meteorology and air quality (e.g., related to the Indian Ocean Experiment, INDOEX) are mainly limited to the South Asian and Indian region (e.g. Kumar et al., 2012a, b; Lawrence and Lelieveld, 2010, and references therein) but only very few model studies have been conducted so far over Nepal or the Kathmandu Valley (e.g. Panday et al., 2009).

Meteorology as well as emissions, mixing and transport, chemistry and deposition of air pollutants are key processes for
air quality. All of these processes are particularly challenging to simulate in the Nepal region because of the very complex topography of the Himalayas and the lack of a dense measurement network, translating into large uncertainties in the lateral boundary conditions from reanalysis data for this region as well as large uncertainties in the parameterized processes in the WRF-Chem model. It is therefore important to ensure a reasonable skill of the model in reproducing the observed meteorology as a precondition for using the model for air quality studies, e.g., assessments of different emission scenarios.

In a first step, a nested model simulation with the WRF model (meteorology only) is performed over South Asia and Nepal, for the time period January through June 2013. This model simulation is then evaluated against available meteorological obser- vations, focusing on the Kathmandu Valley and on the temporal and spatial distribution of meteorological parameters that are particularly relevant to air quality such as, for instance, temperature, wind speed and direction, mixing layer height and pre- cipitation. In a second step, two WRF-Chem simulations including chemistry and aerosols are analyzed with a particular focus
on black carbon concentrations in the Kathmandu Valley. The first WRF-Chem simulation uses data from the readily available emission database EDGAR HTAP v2.2; in the second simulation, the black carbon emission fluxes for the valley are modified to be consistent with a top-down emissions estimate based on SusKat-ABC measurements of black carbon concentrations and mixing height layer in the valley (Mues et al., 2017). Both WRF-Chem simulations are performed for two different months (February and May 2013) representing different meteorological regimes, the dry winter season and the pre-monsoon season.
The black carbon concentrations from both WRF-Chem simulations are evaluated against measurements and compared against each other in order to assess the skill of the model in reproducing observed black carbon levels and the possibility to improve available emission data that are known to have a large uncertainty in this region.

The WRF model and the WRF-Chem model have been widely used for a variety of different applications and have been evaluated against observations in different regions, including, for instance, Europe (e.g. Tuccella et al., 2012), North America
(e.g. Yver et al., 2013) and East Asia (e.g. Gao et al., 2014). Kumar et al. (2012a) set up the WRF-Chem model over South



Asia and evaluated the simulated meteorological fields for the year 2008 against observations. They found that the spatial and temporal variability in meteorological fields is simulated well by the model, with temperature and dew point temperature being typically overestimated during the summer monsoon and underestimated in winter. They also found that the spatio-temporal variability of precipitation is reproduced reasonably well in this region but with an overestimation of precipitation in summer and an underestimation during other seasons. In the literature reviewed for this study, black carbon concentrations are consistently underestimated by the WRF-Chem model, independent of the region (e.g., Europe (Tuccella et al., 2012), East Asia (Zhang et al., 2016) and South Africa (Kuik et al., 2015)).

## 2 Model description, model simulations, observational data, and evaluation metrics

### 2.1 The WRF/WRF-Chem model and model simulations

The Weather Research and Forecasting Model (WRF) model is a widely used three dimensional atmospheric model that offers a large set of physical parameterizations including multiple dynamical cores. WRF is a community model and has been developed through a collaborative partnership of numerous agencies with main contributions from the National Center for Atmospheric Research (NCAR) and NOAA's National Centers for Environmental Prediction (NCEP). WRF can be applied at different horizontal and vertical resolutions and over different regions. The option of nested simulations allows for high-resolution simulations at, for instance, 3 km over a domain of particular interest. WRF-Chem is an extended version of WRF including atmospheric chemistry and aerosols. WRF-Chem can simulate trace gases and particles in an interactive way allowing for feedbacks between the meteorology and radiatively active gases and particles.

In this study WRF and WRF-Chem version 3.5.1 are used. In WRF-Chem we apply the RADM2 chemistry scheme with the MADE/SORGAM aerosol module and aqueous phase chemistry (CMAQ). The combination of RADM2 and MADE has already been applied in many different studies (e.g. Grell et al., 2011). Aqueous phase chemistry has been switched on as we expect this to be of relevance particularly when simulating aerosols and their wet deposition during the pre-monsoon season. The model domain (D01) covers large parts of the Himalayas, India and Nepal (68-107°E, 16-43°N, Fig. 1a) at a horizontal resolution of 15 x 15 $km^2$. The central part of Nepal and the Kathmandu Valley are covered by an additional nested domain (D02) at a horizontal resolution of 3 x 3 $km^2$ (Fig. 1b). WRF and WRF-Chem are configured with 31 vertical $\sigma$-levels and with a model top at 10 hPa. The complete set of physics and chemistry options as well as the data used as initial and lateral boundary conditions and emissions used are summarized in Tab. 1.

Two modifications have been applied to WRF-Chem compared to the standard model version. Firstly, the online calculation of the sea salt emissions in the default WRF-Chem version does not distinguish between ocean and freshwater grid cells (lakes). The model code has been modified to prevent sea salt emissions from small in-land lakes. Secondly, currently there is no calculation of gravitational settling of aerosol particles in WRF-Chem for the chemical mechanism used in this study. Gravitational settling of particulate matter following the method implemented for aerosol particles in the Goddard Chemistry Aerosol Radiation and Transport (GOCART) model (Ginoux et al., 2001) but using the sedimentation velocities calculated by the aerosol module MADE has been implemented into the model code.



The model configuration was tested in several sensitivity simulations to find the "best" combination for the study region, and are chosen in such a way to allow for simulations over a time period of six months and over a relatively large area, and to use the same model setup for the WRF-Chem simulations. Certain aerosol and chemistry options in WRF-Chem are compatible with only specific physics options. Therefore the physics options for the meteorology only simulation (WRF) have been chosen

in such a way that they are compatible with the chemistry and aerosol scheme in the WRF-Chem simulations.

The main characteristics and the acronyms of the WRF and WRF-Chem simulations analyzed in this study are summarized in Tab. 2. The reference simulation WRF_ref is a one-way nested meteorology only (WRF) simulation with two domains (WRF_ref_D01, WRF_ref_D02) (Fig. 1). The time period January through June 2013 has been chosen to cover the entire measurement period of the SusKat-ABC campaign providing a comprehensive set of meteorological and air pollutant measure-

ments that are well suited for comparison with the model results. Two different nested model simulations have been performed with WRF-Chem (including chemistry and aerosols) for the months February and May 2013. The month of February has been chosen as an example of a month in the dry season and because the brick kilns, which are in operation then, are thought to be major emitters of black carbon in the Kathmandu Valley. The brick kilns are typically active between December and April and generally emit continuously throughout the entire day and night. In contrast, May represents a month in the transition phase to

the monsoon season (summer) and other sources with more pronounced diurnal cycles become main emitters of black carbon. The first WRF-Chem simulation (WRFchem_ref) has been performed using the global EDGAR HTAP emission inventory v2.2 which is described in more detail in section 2.2.1. For the second WRF-Chem simulation (WRFchem_BC) the EDGAR HTAP emission inventory v2.2 has also been used, but with the black carbon emission values inside the Kathmandu Valley modified to be consistent with estimates based on measurements of black carbon concentrations and mixing layer height (Mues et al.,

2017). A detailed description of the emission flux estimates is presented in the section 2.2.2.

## 2.2 Black carbon emission data

### 2.2.1 EDGAR HTAP

The gridded EDGAR HTAP v2.2 air pollutant emission data (Janssens-Maenhout et al., 2015) combine the latest available regional information within a complete global data set (EDGAR: Emission Database for Global Atmospheric Research of the

Joint Research Centre, JRC, of the European Commission, in cooperation with the Task Force on Hemispheric Transport of Air Pollution, TF HTAP, organized by the United Nations Economic Commission for Europe's Convention on Long-range Transboundary Air Pollution, LRTAP). HTAP uses nationally reported emissions combined with regional inventories. The emission data are complemented with EDGAR v4.3 data for those regions with missing data. The global data set is a joint effort of the U.S. Environmental Protection Agency (US-EPA), the MICS-Asia group, EMEP/TNO, the REAS and the EDGAR

group for scientific studies of hemispheric transport of air pollution. The EDGAR HTAP v2.2 data set provides emissions of $CH_4$, CO, $SO_2$, $NO_x$, NMVOC, $NH_3$, PM10, PM2.5, BC and OC on a $1° \times 1°$ grid for the years 2008 and 2010 with a monthly time resolution. In the region considered in this study the emissions are based on data from the Regional Emission inventory in Asia (REAS) (Kurokawa et al., 2013), which has a resolution of $0.25° \times 0.25°$.



### 2.2.2 Observational-based estimates of black carbon emission fluxes for the Kathmandu Valley

In Mues et al. (2017) a method is presented to estimate black carbon emission fluxes for the Kathmandu Valley from mixing
layer height data, derived from ceilometer measurements, and black carbon concentrations measured during SusKat-ABC at
the Bode station (number 0017) located within the valley (Tab. 3 and Fig. 1). These estimated emission fluxes are based on
measurement data from March 2013 to February 2014 and calculated for each month. The emission estimates are based on the
assumptions that (i) black carbon aerosols are horizontally and vertically well mixed within the mixing layer, (ii) the variation
of the mixing layer height is only small at night (as frequently observed in the ceilometer measurements used in the study), (iii)
the vertical mixing between the mixing layer and the free atmosphere is small (consistent with a stable mixing layer height),
and (iv) the horizontal transport of air pollutants into and out of the valley is small (consistent with low nocturnal wind speeds).

The use of these observationally-based black carbon emission fluxes is motivated by the finding that the emission fluxes in
the EDGAR HTAP inventory for the Kathmandu Valley are rather small compared to other big cities such as Delhi and Mumbai,
where black carbon concentrations are measured that are similar to the black carbon measurements in the Kathmandu Valley.
Table 4 summarizes the main differences between the two emission data sets for the Kathmandu Valley for February and May.
In the simulation WRFchem_BC these monthly means were used as black carbon emission fluxes for the grid cells representing
the valley. For all other grid cells the EDGAR HTAP emissions are used. For a more detailed description of the estimation of
the black carbon emission fluxes we refer to Mues et al. (2017).

### 2.3 Observational data

Measurements of several meteorological parameters and black carbon concentrations are used in this study to evaluate the
model performance. These measurements were collected from different sources. An overview of the locations of the measure-
ment stations is presented in Fig. 1 and Tab. 3, more details on the sources of the measurements are given below.

### 2.3.1 SusKat-ABC field campaign

The SusKat project started with a two months long intensive measurement campaign (December 2012 to February 2013), which
was extended until June 2013 providing detailed observations of a large number of chemical compounds and meteorological
parameters. From December 2012 to June 2013 more than 40 scientists representing nine countries and 18 research groups
deployed more than 160 measurement instruments for intensive ground-based monitoring at the urban supersite Bode and
a network of 22 additional satellite and regional sites in the Kathmandu Valley and other parts of Nepal (Rupakheti et al.,
2017). SusKat-ABC was so far the second largest international air pollution measurement campaign conducted in South Asia,
following the Indian Ocean Experiment during 1998 to 1999 (Ramanathan et al., 2001; Lelieveld et al., 2001). SusKat-ABC
provides the most detailed air pollution data for the foothills of the central Himalayan region available to date. Hourly data of the
following meteorological parameters are available: near-surface temperature, wind direction and speed, relative humidity and
precipitation. Furthermore, data on the mixing layer height derived from ceilometer measurements are available (Mues et al.,
2017). Black carbon measurements at the Bode site are used in this study for comparison with the WRF-Chem simulations.





The black carbon concentrations were measured with a dual-spot Aethalometer (Aethalometer AE33, Magee Scientific, USA) (Drinovec et al., 2015) with a time resolution of one minute. For the model evaluation, all data are used with a time resolution

of one hour calculated as means from the original data. In contrast to the densely built-up center of the Kathmandu Valley, the surroundings of the Bode site are characterized by a mixed residential and agricultural setting in a suburban location with only light traffic and scattered buildings.

### 2.3.2 DHM measurement data

The Department of Hydrology and Meteorology (DHM) of the Ministry of Population and Environment of the Government of

Nepal hosts a network of meteorological stations. Data from five stations within this network were used in order to compare the meteorology simulated with WRF to observations. Hourly data of 2m temperature and 10m wind speed and direction were used (Tab. 3).

### 2.3.3 ERA-Interim dataset

ERA-Interim is a reanalysis dataset compiled by the European Centre for Medium-Range Weather Forecasts (Dee et al., 2011).

Zonal and meridional wind fields at 500 hPa are used for comparison with the modeled wind fields, as a general consistency check of the model results. As observations in this region are scarce, the reanalysis data for this region is expected to have larger uncertainties than in regions with a higher coverage of observations.

### 2.3.4 Radiosonde data

No radiosonde data are available for the Kathmandu Valley, but radiosonde data from the Integrated Global Radiosonde Archive

(IGRA) at two locations (Tab. 3) within the modeling domain D01 can be used for comparison with the model results (Durre et al., 2006, 2008; Durre and Yin, 2008). Both of these two radiosonde stations are located in northern India (Fig. 1), and only one of the stations lies within the highly resolved model domain D02. For station 42182 (New Delhi/Safdarjung), observations are available at around 00 UTC and 12 UTC between January and June 2013. For station 42379 (Gorakhpur), observations are available only at around 00 UTC. The processing of the radiosonde observations is further described in section 2.4.

### 2.3.5 Tropical Rainfall Measuring Mission (TRMM) data

TRMM based precipitation estimates are used to analyze the geographical distribution of the simulated precipitation fields (Adler et al., 2000). TRMM is a joint mission of NASA and the Japan Aerospace Exploration Agency (JAXA) to measure tropical rainfall for weather and climate research. The TRMM precipitation data are widely used and contributed to improving the understanding of, for instance, tropical cyclone structure and evolution, convective system properties, lightning-storm relationships, climate and weather modeling, and human impacts on rainfall. For the analysis in this study daily precipitation rates with a spatial resolution of 0.25° x 0.25° were used (TRMM product 3B-42).





## 2.4 Evaluation metrics

The model setup chosen in this study is particularly aimed at performing air quality studies in the Kathmandu region. Therefore, a focus in the evaluation of the WRF simulation is on meteorological parameters which are particularly important for air quality. This includes the meteorological parameters temperature, wind speed and direction, the mixing layer height and precipitation. A special focus of the evaluation is on measurement stations in the valley because suitable air quality measurements are only available for this region. For this reason, in particular results for the nested second domain (D02) are shown and discussed.

In order to analyze the performance of the WRF model over the target region, the WRF simulation is compared against measurements obtained at surface stations, from radiosondes, as well as satellite products (see section 2.3). For the comparison with the gridded observational data (ERA-Interim) the model results were interpolated onto a regular longitude - latitude grid applying a simple inverse distance square weighting method. In case of the station measurements a station-to-model-grid comparison is done, meaning that the simulation results from the grid cell in which the individual station is located, are

compared to the station measurements. The model results were output every three hours starting at 00 UTC. For the model evaluation only hours with both model and measurement data available, were taken into account when producing the figures and the statistics. Here, stations are only considered when they have a data availability of at least 70 % based on hourly data for the time period of interest (except for the mixing layer height) (Tab. 3).

     Radiosonde data are compared to model results in order to evaluate the model's skill in reproducing the observed vertical

structure of the atmosphere. Both the observations and model data are averaged over the same pressure bins as well as over the whole period of six months. The mean temperature and the median relative humidity over the whole time period and each pressure bin are compared here. The standard deviation indicates the variability over the whole time period within each bin. For station "42182", observations were available at around 00 UTC and 12 UTC. As launch time of the radiosondes varied, observations for 00 UTC also include 23 UTC and 01 UTC observations, and profiles for 12 UTC also include observations

for 11 UTC, 13 UTC and 14 UTC. In total, 174 profiles were available at around 12 UTC and 180 profiles were available at around 00 UTC. For station "42379", observations were available only at around 00 UTC, which also includes observations at 01 UTC and 02 UTC due to varying launch times. In total, 77 profiles were available. Model results have only been included if observations exist for the respective times.

     The statistical metrics used to evaluate the model performance are mean bias (MB) (Eq. 1), root mean square error (RMSE)

(Eq. 2) and the Pearson (temporal) correlation coefficient (r) (Eq. 3). The metrics are defined as follows, with N being the number of model and observation pairs, M the model and O the observation values and $\sigma_M$ and $\sigma_O$ the standard deviations of modeled and observed values, respectively:

$$MB = \frac{1}{N}\sum_{i=1}^{N}(M_i - O_i) \qquad (1)$$

$$RMSE = \sqrt{\frac{\sum_{i=1}^{N}(M_i - O_i)^2}{N}} \qquad (2)$$





$$r = \frac{1}{N-1} \sum_{i=1}^{N} \left( \frac{M_i - \overline{M}}{\sigma_M} \right) \left( \frac{O_i - \overline{O}}{\sigma_O} \right) \tag{3}$$

The precipitation simulated by the model is evaluated against measurements taken at the Bode site and against daily precipitation fields from TRMM (see section 2.3.5). The TRMM data are averaged over domain D02 as an estimate for the precipitation particularly relevant to air pollutant concentrations in the Kathmandu Valley and its surroundings. In the context of air quality, a good hit rate of the occurrence of precipitation events by the model is especially important, rather than the exact representation of the amount of precipitation. The hit rate (H) (Eq. 4), the false-alarm ratio (FAR) (Eq. 5) and the critical success index (CSI) (Eq. 6) (Kang et al., 2007) have been calculated for precipitation at the Bode site and the time period January to June 2013. These metrics are calculated as followed:

$$H = \left( \frac{b}{b+d} \right) \cdot 100\% \tag{4}$$

$$FAR = \left( \frac{a}{a+b} \right) \cdot 100\% \tag{5}$$

$$CSI = \left( \frac{b}{a+b+c} \right) \cdot 100\% \tag{6}$$

Here, a represents the number of forecast precipitation days (daily sum >0.5 mm) that were not observed, b represents the number of correctly forecast precipitation days, d represents the number of precipitation days which were not forecast. Metric H is the percentage of observed precipitation days that is correctly forecast by the model. CSI indicates how well precipitation days were predicted by the model by considering false alarms as well as missed forecasts of precipitation days. In order to compare the two different observations (station measurements and TRMM data) the metrics have also been calculated for the satellite data.

## 3 Results

### 3.1 Evaluation of the WRF model simulation - Meteorology

#### 3.1.1 Zonal and meridional wind fields

As a first assessment of the model's performance in reproducing the large-scale wind pattern, the model results are compared to the 500 hPa wind fields from the ERA-Interim reanalysis. It should be kept in mind that because of the sparsity of available observations in this region, the reanalysis data for this region is expected to have larger uncertainties than in better observed



regions. The spatial distribution of the zonal and meridional wind components at 500 hPa from WRF and the ERA-Interim
reanalysis averaged over February and May 2013 are shown in Fig. 2. The overall pattern of the zonal wind component is
qualitatively similar in both data sets for February, with lower values over India in the model simulation. Differences of up to 5
$\mathrm{m\,s^{-1}}$ are found in the zonal wind component in February south of the Himalayas extending in east-west direction throughout
the whole model domain. In May, the zonal wind speed at 500 hPa simulated with the model is much lower compared to ERA-
Interim data as shown by the domain averaged mean bias of 2.9 $\mathrm{m\,s^{-1}}$. ERA-Interim shows here a stronger westerly wind
component. The spatial distribution of the meridional wind component simulated by the model is also qualitatively similar to
the ERA-Interim fields in both months, with some difference in the southeast of domain D01 in February and over India in
May 2013. The domain averaged mean bias of the monthly mean meridional (zonal) wind fields is 0.1 $\mathrm{m\,s^{-1}}$ (2.2 $\mathrm{m\,s^{-1}}$) for
February and 0.3 $\mathrm{m\,s^{-1}}$ (2.9 $\mathrm{m\,s^{-1}}$) for May and the spatial correlation of the meridional and zonal wind distributions are
0.9/0.8 and 0.9/0.8 for February and May, respectively.

### 3.1.2   Vertical profiles

In order to evaluate the ability of the model to correctly represent the vertical structure of the atmosphere, measurements from
radiosondes for temperature and relative humidity are compared to the model results (Fig. 3 and 4). This comparison only
provides a limited quality check of the model, since there is only one single radiosonde station available within D02. The
comparison shows that WRF is able to capture the basic features of the vertical profiles of temperature and relative humidity
with the modeled vertical profiles being within the variability estimated by the standard deviation (shaded areas), with the
largest differences typically between about 900 and 700 hPa and near the surface.

### 3.1.3   2m temperature

The daily mean 2m temperature increases during the simulation period at all stations shown in Fig. 5, from about 5 - 10 °C in
January to 20 - 30 °C in June which is also shown by the model (WRF_ref_D02). While the observed temporal evolution of
the daily mean near-surface temperature is well reproduced by the model (correlation above 0.9 (Fig. 5)), the absolute values
are systematically over- or underestimated at several stations. The mean bias for WRF_ref_D02 ranges between -1.9 and 2.2
K (Fig. 5). At several stations the over- or underestimation of measured temperature is associated with a difference between
the actual elevation of the measurement station and the elevation of the model grid cell the station is located in. For example,
at station "1206", the elevation of the grid cell in the domain D02 is 149 m lower than the elevation of the measurement station
(1720 m); given a typical atmospheric vertical temperature gradient of 6 - 7 $\mathrm{K\,km^{-1}}$, one would expect a bias of about 1 K,
which is close to the actual mean temperature bias of 0.8 K. In order to correct for the temperature biases caused by differences
in elevation, a height correction has been applied to the model data by linearly interpolating the modeled vertical temperature
profile to the elevation of the measurement station. For the stations the mean bias reduced by 1 K ("0014") to 0.2 K ("1206")
(Fig. 5) when considering this height correction. Table 5 summarizes the statistics averaged over all available stations and the
whole simulated time period based on 3-hourly data. On average, the model overestimates the observed mean temperatures
by 0.7 K. The mean daily minimum and maximum temperatures are overestimated by 1 K and underestimated by 0.5 K,





respectively. The main features of the average diurnal cycle of the 2m temperature (Fig. 6) are reproduced by the model but
the daily temperature amplitude (difference between the daily minimum and maximum temperature) are often smaller in the
model simulation than in the measurements. This is mainly caused by a high bias in the simulated values in the morning hours.
In contrast, the daily variability of the 2m temperature shown by the 25th and 75th percentiles in Fig. 6 is reproduced quite
well by the model.

The temperature biases found at stations in the present study are in the same range as the ones found in other regions
with WRF (Zhang et al., 2013, 2016; Mar et al., 2016; Kuik et al., 2015), particularly when considering that the reported 2m
temperature biases in these studies tend to be higher in mountainous terrain than in other regions. For example, Zhang et al.
(2016) found a mean bias in the 2m temperature of -1.5 to 1 K at stations in East Asia, while at single stations the mean bias
can range between -5 and +5 K in January and July 2005, respectively. Kuik et al. (2015) found a good agreement between
WRF-Chem simulations for South Africa and ERA-Interim reanalysis data 2m temperature in 2010 (mean bias 0.4 K and
-0.03 K, spatial correlation 0.93 and 0.91, for September and December, respectively). Mar et al. (2016) found that the spatial
variability in measured 2m temperature is well reproduced by WRF-Chem in all seasons in 2007 over Europe with values of
the absolute mean bias of generally less than 1 K. Both Mar et al. (2016) and Zhang et al. (2013) found the largest biases in
2m temperature in the Alps. Mar et al. (2016) describes an overprediction by more than 1 K in this region whereas Zhang et al.
(2013) found a cold bias of -5 to -2 K.

### 3.1.4 10m wind speed and direction

The wind speed has an essential impact on the horizontal transport of pollutants. For example, low wind speeds favor an
accumulation of pollutants close to their sources whereas higher wind speeds lead to the transport of pollutants away from
their source. The average measured wind speed over all stations and over the six months based on hourly data is $1.7\,\mathrm{m\,s^{-1}}$
(Tab. 5), which is overestimated by the model by $1\,\mathrm{m\,s^{-1}}$. At individual stations where wind speed data is available the biases
ranges between 0 and $1.7\,\mathrm{m\,s^{-1}}$. The temporal correlation coefficient of hourly wind speed is on average 0.4 with a range of
0.1 to 0.6 at these individual stations (Tab. 5). The overestimation in wind speed in the WRF_ref_D02 simulation can probably
be attributed to a large extent to an overestimation of the maximum wind speed during daytime, which is on average biased
positively by $2\,\mathrm{m\,s^{-1}}$. In contrast, the daily minimum wind speed is close to the observation (MB of $0.2\,\mathrm{m\,s^{-1}}$) (Tab. 5). This is
also clearly seen in the frequency distributions of the wind speeds (Fig. S1), which typically have a much broader distribution
with higher wind speeds and a maximum shifted to larger values for the model compared to the observations.

This performance of WRF in reproducing the observed mean 10m wind speed is consistent with biases reported in the
literature, especially when considering stations in mountain regions. For example, Mar et al. (2016) found an overestimation of
the modeled wind speed over Europe, especially during winter and fall with a bias of $2\,\mathrm{m\,s^{-1}}$ and more. Regions with a larger
bias include the mountain region of the Alps, indicating the challenges of simulating wind accurately over complex terrain. The
temporal correlation of the modeled 10m wind speed in Europe is typically above 0.7, but lower (0.4 - 0.6) over the Alps and
close to the Mediterranean (Mar et al., 2016), which is still higher than found at some stations in this study. Zhang et al. (2013)
describe a significant overprediction at almost all sites investigated in Europe (MB of $2.1\,\mathrm{m\,s^{-1}}$) with the largest biases over



several countries in low-lying coastal areas and over the Alps as well as the Carpathian Mountains. They argue that these results

indicate the difficulty of the WRF model in simulating wind patterns and mesoscale circulation systems (such as sea breeze and bay breeze) and their interaction with land over complex terrain. Furthermore, they state that this high bias in 10m wind speed can be mainly attributed to a poor representation of surface drag exerted by the unresolved topography in WRF. Yver et al. (2013) tested different planetary boundary layer (PBL) schemes in their model setup and also found an overestimation of wind speed at stations in California in all cases, although of different magnitude (about 0.5 to 3 $\mathrm{m\,s^{-1}}$). Zhang et al. (2016)

found a significant overprediction of 10m wind speed at stations in East Asia with a mean bias of 1.9 - 3.1 $\mathrm{m\,s^{-1}}$.

An evaluation of the 10m wind speed and especially the wind direction at the individual measurement stations (not shown) strongly suggests that these parameters are highly dependent on the stations' locations and the topography of their surroundings, especially in mountain areas. The measurements at some of these sites are therefore probably only representative for a rather small area around the station. Because of the complex topography in this region, a horizontal resolution of 3 x 3 $\mathrm{km^2}$ is too

coarse to represent the near-surface wind at sites strongly influenced by small-scale features such as individual mountains. Therefore, the main focus of the evaluation of the 10m wind is on the Kathmandu Valley. The Kathmandu Valley with a diameter of about 30 $\mathrm{km}$ is starting to be large enough to be resolved at the model resolution of 3 x 3 $\mathrm{km^2}$. The relatively flat valley floor further facilitates a comparison of the 3 x 3 $\mathrm{km^2}$ model grid cells with observational data as measurements inside the valley are expected to be less influenced by small-scale topography than at most stations outside the valley.

The frequency distribution of wind speed per wind direction based on 3-hourly data for the whole simulation period is shown in Fig. 7 as wind roses for all available stations in the valley. The main wind directions in the east of the valley (station 1015) are north northwest, east southeast and south, with wind speeds of typically up to 6 $\mathrm{m\,s^{-1}}$. Different to the observations the model shows wind directions from north northwest to south southeast. Wind speeds are similar as observed. The main wind direction at stations in the west of the valley (0014 and 0017) is less clearly dominated by particular sectors than in the east

of the valley but rather characterized by predominately westerly winds. This pattern is reproduced by the model although the wind speed is generally overestimated. The observed diurnal cycle of wind speed at the Bode station (Fig. 8a) shows very low median values between 0 and 1 $\mathrm{m\,s^{-1}}$ during the night and a maximum median wind speed during daytime of about 4 $\mathrm{m\,s^{-1}}$. As discussed before, the low wind speed during night is well reproduced by the model but the maximum wind speed during daytime is overestimated. The main wind direction during night time is from the east southeast (around 100°) in the

observations (Fig. 8b), while it is from ca 180°in the model. For such low wind speeds, however, the measured wind direction is expected to be affected by small-scale dynamics such as turbulence and thus not expected to be directly comparable to a 3 x 3 $\mathrm{km^2}$ model grid cell. In the transition phase from low to high wind speed during morning hours (9 - 11 LT) and from high to low wind speed in the evening (19 - 21 LT) the model does not reproduce the wind direction correctly. In contrast, the main wind direction during daytime is west-south-west (around 250°) which is reasonably well reproduced by the model.

### 3.1.5   Mixing layer height

A key parameter for air quality is the depth of the mixing layer which is a part of the planetary boundary layer and characterized by a strong gradient in parameters such as potential temperature and aerosol concentration and by an unstable layer and strong





mixing due to turbulence during daytime and a rather stable layer during night time. Thus, the mixing layer has an important

impact on the dispersion or accumulation of pollutants at the ground level. In the WRF model the mixing layer height is a diagnostic variable which is calculated based on the Richardson number (Hong et al., 2006). The model output is compared to the values derived from ceilometer measurements obtained during SusKat-ABC (Mues et al., 2017). In Fig. 9 the diurnal cycle of the mixing layer height calculated from data covering the time period January to June 2013 is shown for the model (WRF_ref_D02) in comparison with the ceilometer data. Both model and observations show a distinct diurnal cycle with low

mixing layer heights during the night and morning hours and higher values during the day. While the lowest measured nocturnal values are around 200 m, the modeled values typically go down to less than 50 m. The maximum mixing layer height values are measured at around 16 LT in the afternoon with a median of 1100 m. The simulated values are higher during the day, with a median of 1200 m at 15 LT. This over- and underestimation of the maximum and minimum in the diurnal cycle are also shown for individual months, for instance, a high/low bias for the maximum/minimum mixing layer height of +244/-76

m in February and +280/-122 m in June. A similar pattern was also found by Kuik et al. (2016) for WRF-Chem simulations over Germany in summer, with a mean bias of -113 m for the daily minimum and 287 m for the daily maximum mixing layer height. Furthermore, the simulated diurnal cycle of the increase in mixing layer height during daytime is shifted by about 2 hours to earlier times compared to the measurements. During the day, convection is an important process for determining the mixing layer height. A premature onset of convection found in many models is a long-standing issue and has been identified in

numerous previous modeling studies, including studies with WRF (e.g. Pohl et al., 2014).

### 3.1.6 Precipitation

A good representation of the precipitation in the model is important for the calculation of wet deposition of air pollutants such as particulate matter including black carbon. The domain averaged daily precipitation totals from the model (WRF_ref_D02) and TRMM are shown as a time series in Fig. 10. The near-absence of strong rain events in the dry season (January through

April) is reproduced well by the model, and also the timing of the single rain events between January and March are reproduced well, although the total amount of precipitation is overestimated by the model. The transition to and start of the rainy season in late April / early May as seen in the TRMM data is also reproduced reasonably well by the WRF simulation.

The statistics summarized in Tab. 6 represents the skill of the model (WRF_ref_D02) to reproduce precipitation events at one single station in the valley (Bode). It shows that 62 / 57 % (H) of the observed precipitations days are correctly captured

by the model when using the Bode station measurements and the TRMM data, respectively, as reference data. The ratio of days when precipitation was present in the model data but not measured relative to all forecasted precipitation days (FAR) is relatively high, 32 % for the station measurements and 36 % for the TRMM data. Other than the hit rate the CSI also considers false alarm and missed forecast, but it is not influenced by correctly forecast no precipitation days. The CSI score indicates that 48 % of the forecast and observed precipitation days are correct. When using the TRMM data as observational reference, the score is a bit smaller (43 %). Hit rate and CSI are both lower for the model if considering TRMM as reference. Differences between the two observational data sets (station measurement and TRMM data) are shown in Tab. 6. The hit rate for the station measurements and the TRMM data (station measurement / TRMM) indicates that 71 % of the measured precipitation days





at the Bode station are also visible in the TRMM data. The differences obtained when using the two different observational

datasets also show the uncertainties and limitations particularly of the TRMM data for this kind of comparison. Since some of the precipitation events can be rather localized (e.g. convective rain) and can thus not be expected to be fully reproduced by a 3 x 3 $\mathrm{km}^2$ model simulation, they might also be missed in the rather coarse spatial and temporal (satellite overpass times) resolution satellite data.

## 3.2   WRF-Chem model simulations of black carbon

### 3.2.1   Results from the WRFchem_ref and WRFchem_BC model simulations

Two WRF-Chem simulations have been performed with an identical model configuration but using different black carbon emissions. The WRF-Chem reference simulation uses the EDGAR HTAP emissions (WRFchem_ref), the second simulation uses the same emission data but with black carbon emission fluxes over the Kathmandu Valley replaced by emission estimates based on SusKat-ABC measurements (WRFchem_BC) (see section 2.2.1 and 2.2.2 for details on the emission data sets). The

black carbon emission fluxes used in both WRF-Chem simulations are shown in Fig. 11.

Monthly mean black carbon concentrations measured in the Kathmandu Valley at the Bode station are 27 $\mathrm{\mu g\,m}^{-3}$ in February 2013 and 11 $\mathrm{\mu g\,m}^{-3}$ in May 2013. These values are strongly underestimated in the reference simulation WRFchem_ref_D02 (using EDGAR HTAP emissions), which average only 3 $\mathrm{\mu g\,m}^{-3}$ (89 % underestimate) in February and 2 $\mathrm{\mu g\,m}^{-3}$ (82 % underestimate) in May. The WRF-Chem sensitivity simulation using the black carbon emission fluxes inside the Kathmandu

Valley estimated from observations (WRFchem_BC_D02) shows significantly reduced biases, averaging 12.5 $\mathrm{\mu g\,m}^{-3}$ (54 % low bias) in February and 6 $\mathrm{\mu g\,m}^{-3}$ (45 % low bias) in May. These results from WRFchem_BC_D02 are in much better agreement with the measurements at the Bode site, even though black carbon is still underestimated by the model. The improvement of the simulated black carbon concentrations when using the observationally-based estimated fluxes can also be seen in the time series of daily mean black carbon concentrations (Fig. 12). Measured daily black carbon concentrations reach values of

up to 35 $\mathrm{\mu g\,m}^{-3}$ in February and up to 28 $\mathrm{\mu g\,m}^{-3}$ in May, with a pronounced variability within the same month (e.g., 2-5 May vs. 6-8 May). The daily mean black carbon concentrations from the reference simulation WRFchem_ref_D02 are below 5 $\mathrm{\mu g\,m}^{-3}$ in both months. The differences between the two months as well as the large daily variability are not reproduced by the reference simulation. In contrast, the time series of the WRFchem_BC_D02 sensitivity simulation shows values of up to 20 $\mathrm{\mu g\,m}^{-3}$ in February and up to 8 $\mathrm{\mu g\,m}^{-3}$ in May. In addition, the observed differences between February and May as well as

the daily variability are better reproduced than in the reference simulation WRFchem_ref_D02. In order to compare the spatial variability of the simulated black carbon concentration in the valley, also the daily mean concentrations simulated in the grid cells with the highest and lowest values of all neighboring grid cells of the "Bode" grid cell are shown in Fig. 12. The spatial variability of the simulated black carbon concentration is higher (in absolute and in relative terms) in the WRFchem_BC_D02 simulation compared to WRFchem_ref_D02. This figure also show that the grid cell with the Bode station is not an outlier but generally at the upper end of the range of minimum and maximum concentrations of its neighbors.



The histogram of the measured hourly black carbon concentrations (Fig. 13) shows values of up to 90 $\mu g\,m^{-3}$ and a maximum of the distribution between 0 and 10 $\mu g\,m^{-3}$. These values of the measured frequency distribution are not reproduced by

the reference simulation WRFchem_ref_D02, in which the black carbon concentrations range only between 0 and 6 $\mu g\,m^{-3}$ with a maximum frequency between 1 and 1.5 $\mu g\,m^{-3}$. The histograms of the WRFchem_BC_D02 simulation for February and May show a wider frequency distribution compared to the reference simulation WRFchem_ref_D02 with maximum concentrations of up to 40 and 20 $\mu g\,m^{-3}$ and maximum frequencies in the interval 0 to 10 $\mu g\,m^{-3}$ and around 5 $\mu g\,m^{-3}$ (in February and May, respectively).

The pollution roses in Fig. 14 show the measured and simulated black carbon concentrations coinciding with each specific wind direction at the Bode station and the frequency of the occurrence of the corresponding wind direction in percent. The figure shows that the observed main wind direction in February is from the west and west-southwest, but high black carbon concentrations are found for all wind directions. Simulated main wind directions span a wider range than in the observations (west-northwest, southwest and south) but the model reproduces the observation that high black carbon concentrations are

found independent of the actual wind direction. In May the observed main wind direction is from the west (and slightly north and south of west), and the highest concentrations are measured for winds from the north and east-southeast (Fig. 14 d). Again the model does not fully reproduce the main wind directions (here northwest to south) and underestimates black carbon concentrations at all wind directions.

These findings strongly suggest that the EDGAR HTAP emissions of black carbon in the valley are underestimated and

that there is a need for further improvements of the local emissions in the Kathmandu Valley. Despite this improvement in the simulated black carbon concentrations in the Kathmandu Valley when using the black carbon emission fluxes estimated from observations, the measured concentrations are still significantly underestimated by the model.

### 3.2.2 Discussion of the observation-based emission estimates for black carbon

Two possible reasons for the above mentioned underestimation of the observed black carbon concentrations in the WR-

Fchem_BC_D02 simulation are an overestimation of the dispersion of the black carbon aerosols away from the ground and too small observation-based black carbon emissions estimates. Even though the model tends to overestimate the observed near-surface wind speed, the model bias of about 1 $m\,s^{-1}$ is not expected to be large enough to explain the large differences in simulated and observed black carbon concentrations through an overestimated horizontal dispersion. The observed and simulated mixing layer heights (Fig. 9) are quite similar, suggesting that the model is able to produce a reasonable vertical dis-

persion. Furthermore, particularly at night time the smaller than observed simulated mixing layer height would rather lead to an overestimate of the observed black carbon concentrations by the model. This suggests that biases in the modeled dispersion (horizontal and vertical) alone are unlikely to be able to explain the large differences in modeled and observed black carbon levels. This, in turn, suggests that the top-down emissions determined by Mues et al. (2017) based on the observed black carbon concentrations and mixing layer heights might be underestimated - despite the fact that they are several times as high as the values in the state-of-the-art EDGAR HTAP v2.2 dataset.



There are various possible reasons why the top-down emissions derived from measurements at the Bode station might be underestimated or not fully representative for the entire Kathmandu Valley as assumed in the sensitivity study WRFchem_BC. One main reason is that the Bode station is not located in the urban center. Thus, throughout most of the year, during the months

when the brick kilns near Bode are not operating, several important urban emission sources such as traffic, cooking and open burning of trash might be underestimated due to applying the top-down method to determine the black carbon emission flux based on the semi-urban Bode site data. Future development of high-resolution (e.g., 1 x 1km$^2$) emissions datasets (Sadavarte et al., manuscript in preparation) may help to resolve this possible discrepancy.

The other main possible reason for the top-down emissions to underestimate actual emissions is that the method currently

only considers sources that are active at night, when the mixing layer height is stable and the increase in black carbon concentrations can be directly attributed to emissions during that time period. It is assumed that the average emissions during the rest of the day are the same as during this period. This can lead to either an over- or underestimation, depending especially on the extent to which the morning food preparation and rush hour traffic occur during the period of the stable nocturnal boundary layer. It is possible that the contribution of black carbon sources which are mainly active during daytime, after the nocturnal

boundary layer begins to break up, exceed the night-time emissions. Since the daytime-specific emissions such as rush hours throughout much of the year and the generally heavier daytime traffic are not taken into account by the top-down computation, this could lead to an underestimation in the black carbon emissions fluxes. This is consistent with the statement by Mues et al. (2017) that the top-down emissions estimate is "likely a lower bound" and thus strongly supports the indication of an under-estimation of the values in current emission datasets. Unfortunately, no technique has yet been found to apply the top-down

method for the full diurnal cycle in the situation of the Kathmandu Valley, so it will be left to emissions inventory developers to improve their estimates based on updated emissions factors and activity data for the region, in order to hopefully determine what is missing according to the top-down analysis.

Despite that offset that is apparently due to the emissions, the temporal correlation coefficient between daily data of the WRFchem_BC_D02 results and the Bode observations is relatively high (0.7) in February, while it is much lower (0.2) in May

2013. There are likely two factors that contribute to this difference. Firstly, in May the day-to-day variability of the emission strength from different sources can expected to be higher because brick kilns, which emit relatively constantly throughout the day and night, are no longer running, and emission sources with a much clearer diurnal cycle like cooking, traffic and trash burning take on a greater relative importance. Secondly, the meteorology in May is more difficult to simulate than in February as convective precipitation becomes more frequent. The correct simulation of the occurrence of daily precipitation events is

particularly important in this context. Although the transition from the dry season in winter to the wet season in summer is captured well by the model, there are several days when precipitation was observed and not simulated in the model and the other way around (Tab. 6), which has an important impact on the simulated day to day variability of black carbon. In addition to particles being removed by wet deposition, also certain emission sources such as burning of trash and biomass can be affected by precipitation.



### 3.2.3 Case study: the episodes 2-5 May and 6-8 May 2013

A 4-day episode of particularly high black carbon concentrations ranging from 20 to 28 $\mu g\,m^{-3}$ was observed between 2 and 5 May 2013, with a maximum on 4 May (Fig. 12). In contrast, comparatively low black carbon concentrations of 5-10 $\mu g\,m^{-3}$ were observed between 6 and 8 May. The simulated black carbon values do not show such a strong difference between these

two episodes and remain rather constant throughout both episodes. There are two main reasons for such observed high black carbon concentration episodes: the meteorological situation or particularly high black carbon emissions during this time period (or a combination of both). To examine the first possibility, the 500 $hPa$ geopotential fields for both episodes do not show any significant differences, which suggests that the large-scale synoptic situation is not a main driver of the large difference in black carbon concentrations between the two episodes. Other important meteorological parameters for air quality such as

wind, mixing layer height (Fig. S2) and precipitation are also quite similar during both episodes, with the mixing layer height being even slightly higher during the first episode. The simulated meteorology is also quite similar during both episodes. This suggests that the high black carbon concentration episode might be primarily caused by enhanced emissions during these days. This would also be consistent with the finding that the model does not reproduce this feature, since monthly mean emission fluxes rather than daily fluxes are used in the model. As this is the case for most model simulations, the models will not be able

to reproduce such emission driven episodes.

## 4 Summary and Outlook

An evaluation of the simulated meteorology with the WRF model over South Asia and Nepal with a focus on the Kathmandu Valley for the time period January to June 2013 is presented in this study. The model evaluation is done with a particular focus on meteorological parameters and conditions that are relevant to air quality. The same model setup is then used for simulations

with the WRF model including chemistry and aerosols (WRF-Chem). Two WRF-Chem simulations have been performed: a reference simulation using emissions from the state-of-the-art database EDGAR HTAP v2.2 along with a sensitivity study using modified, observation-based estimates of black carbon emission fluxes for the Kathmandu Valley. The WRF-Chem simulations have been performed for February and May 2013 and are compared to black carbon measurements in the valley obtained during the SusKat-ABC campaign.

The ability of the model to reproduce the large scale circulation is tested in this study by comparing the simulated zonal and meridional wind components on the 500 $hPa$ level to ERA-Interim reanalysis data. The spatial distribution of the simulated wind fields is in good agreement to the ERA-Interim fields except for the zonal wind component in May when large differences between the two datasets are found over the whole domain. WRF is also able to capture the basic features of the vertical profiles of temperature and relative humidity, with the modeled vertical profiles being within the variability of the measurements from

radiosondes in India, although differences are clearly seen in the profiles for relative humidity near the ground. At most of the stations, the modeled 2m temperature is biased positively with an average bias of less than 1 K, which is well within the range of temperature biases found in other WRF studies. The average temporal correlation of the modeled 2m temperature is 0.9. In the 2m temperature diurnal cycles the main features of the cycle are reproduced by the model, but the daily temperature





amplitudes are often underestimated by the model. The measured 10m wind speed and direction are typically highly dependent on the stations' locations and the topography of their surroundings, and thus difficult to compare with a 3 x 3 km$^2$ horizontal model resolution. For wind speed, especially the maxima during daytime are overestimated by the model, which is also found in other WRF studies particularly in mountain areas. The temporal correlation of wind speed is comparably low, highlighting

again the difficulty to represent station measurements of 10m wind speed with this model resolution. In contrast, the wind measurements taken inside the Kathmandu Valley are considered more representative for a larger area such as a model grid cell, as the topography inside the valley is more homogeneous than in the surroundings of the other measurement stations. The wind direction at stations in the Kathmandu Valley is in general reproduced reasonably well considering the generally quite complex topography in the whole model domain. The modeled mixing layer height is compared to ceilometer data obtained at

the Bode station inside the valley and shows a good overall agreement, but with a 10 % overestimation in mixing layer height during daytime and a shift of the diurnal cycle by about 2-3 hours earlier than observed. For precipitation, the transition from the dry to the rainy season is fairly well reproduced by the model, although the amount of precipitation per day is different than in the TRMM data. During the six months about 62 % of observed precipitation days at the Bode station in the valley are correctly captured by the model. In general, the results for most meteorological parameters are well within the range of

biases found in other WRF studies especially in mountain areas. But the evaluation results also clearly highlight the difficulties of capturing meteorological parameters in complex terrain and reproducing subgrid-scale processes. To address these issues a higher horizontal resolution in the model would be necessary, which would then also require a higher resolution of the input data, which are currently not available for this region.

The simulated meteorology has an important impact on the skill of the model in correctly representing air pollutants in

the WRF-Chem simulations. The focus here is on the Kathmandu Valley and black carbon concentrations as a pre-study of assessing different air pollution mitigation scenarios in the future. The overestimation of daytime wind speed and mixing layer height might lead to an overly rapid transport of black carbon away from its sources and out of the valley, and thus to an enhanced effective vertical mixing and too strong dilution of black carbon near the surface. The low wind speeds in the valley during nighttime are reproduced well by the model and thus the resulting accumulation of black carbon at night can in principle

be captured by the model although the underestimation of the nighttime mixing layer height by the model will tend to cause too much accumulation of black carbon at night. Most precipitation and dry days were correctly forecast by the model (a total of 142 days), while 22 precipitation days were not and 17 were incorrectly forecast. On individual days, the incorrect simulation of precipitation can lead to an over- or underestimation of wet deposition of black carbon.

In addition to the meteorology, also a good representation of the emissions is crucial in order to simulate air pollutants

such as black carbon concentrations correctly. Using the state-of-the-art emission database EDGAR HTAP v2.2 in the WRF-Chem simulation leads to a very strong underestimation of the measured black carbon concentration at the Bode station, with a monthly mean bias of about 90 % in February and 80 % in May. Using top-down estimated emission fluxes for black carbon this bias can be reduced to about 50 %. This confirms the strong need for an updated black carbon emission database for this region. However, it also became clear that a simple correction of the emission fluxes using the top-down method

by Mues et al. (2017) also has several limitations. One of these limitations is an over-representation of emissions which are





relatively constant throughout the day (e.g., from brick kilns) while underrepresenting emissions which are mainly occurring during the daytime (e.g., traffic). In addition, the analysis showed that the monthly mean emissions currently used in the model cannot resolve short-term episodes with reduced or enhanced emission fluxes. The analysis of the observations further suggests that such episodes play an important role in explaining the observed variation in daily black carbon concentrations in

the valley. In order to further improve the simulation of black carbon, an updated emission database for the Kathmandu Valley and its surroundings is essential. Emission time profiles, describing the diurnal cycle of emission per sector, especially for months when the continuously emitting brick kilns are not active, are expected to further improve the simulation results. Such improvements of the emission data seem urgently needed before being able to use the model to robustly assess air pollution mitigation scenarios in this region in a meaningful way.

## 5 Code availability

WRF-Chem is an open-source community model. The source code is available at http://www2.mmm.ucar.edu/wrf/users/download/get_source.html. The two modifications described in Sect. 2 are available online via ZENODO at http://doi.org/10.5281/zenodo.1000750.

## 6 Data availability

The initial and lateral boundary conditions used for the model simulations in this study are publicly available. Meteorological fields were obtained from ECMWF at http://www.ecmwf.int/en/research/climate-reanalysis/era-interim/ and chemical fields from MOZART-4/GEOS-5, provided by NCAR at http://www.acom.ucar.edu/wrf-chem/mozart.shtml. Anthropogenic emissions were obtained from EDGAR HTAP available at http://edgar.jrc.ec.europa.eu/htap_v2/. Observational data from TRMM are available from NASA at https://pmm.nasa.gov/data-access/downloads/trmm/, radiosonde data from the Integrated Global

Radiosonde Archive (IGRA) at https://www.ncdc.noaa.gov/data-access/weather-balloon/integrated-global-radiosonde-archive/ and ERA-Interim reanalysis data from ECMWF at http://www.ecmwf.int/en/research/climate-reanalysis/era-interim/. Meteorological data from stations maintained by the Department of Hydrology and Meteorology (DHM), Nepal can be purchased from the DHM, Nepal. SusKat-ABC data will also be made publicly available through the IASS website. SusKat-ABC campaign data used in this study can also be obtained by emailing to the first author.

*Acknowledgements.* We would like to thank the WRF and WRF-Chem developers for their support in setting up the model. We would furthermore like to acknowledge the Department of Hydrology and Meteorology (DHM) of the Ministry of Population and Environment of the Government of Nepal for providing station measurements of meteorological parameters. We acknowledge the National Research Council of Italy (Institute of Atmospheric Sciences and Climate) for elaborating meteorological parameters recorded by Ev-K2-CNR at the Paknajol station. This work was hosted by IASS Potsdam, with financial support provided by the German Research Foundation (DFG), the Federal Ministry of Education and Research of Germany (BMBF) and the Ministry for Science, Research and Culture of the State of Brandenburg

(MWFK).



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



**Table 1.** WRF and WRF-Chem setup including namelist settings.

| WRF/WRF-Chem model setup | Description |
| --- | --- |
| **Model version** | 3.5.1 |
| **Domain** | |
| Domain D01 | Resolution: 15x15 km$^2$ |
| | Latitude: 15.5°- 43.5°, Longitude: 67.6°- 107.4° |
| | Number of grid cells: west-east 221, north-south 201 |
| Domain D02 | Resolution: 3x3 km$^2$ |
| | Latitude: 25.4°- 29.6°, Longitude: 82.6°- 87.9° |
| | Number of grid cells: west-east 171, north-south 151 |
| | One-way nesting |
| Vertical levels | Number of levels: 31 $\sigma$-levels, model top: 10hPa |
| **Physics** | |
| Microphysics Scheme | Lin et al. (option 2) (Lin et al., 1983) |
| Longwave Radiation Scheme | RRTMG (option 4) (Iacono et al., 2008) |
| Shortwave Radiation Scheme | Goddard (option 2) (Chou and Suarez, 1994) |
| PBL Physics Scheme | YSU (option 1) (Hong et al., 2006) |
| Surface Layer | Revised MM5 scheme (option 11) (Jiménez et al., 2012) |
| Cumulus Parametrization Scheme | New Grell (option 5) (Grell, 1993; Grell and Dévényi, 2002) |
| Land Surface Model | Noah land-surface model (option 2) (Tewari et al., 2004) |
| **Chemistry** | |
| Chemistry option | RADM2/SORGAM with aqueous reactions included |
| | feedback between meteorology and chemistry switched on (option 41) |
| | (Ackermann et al., 1998; Schell et al., 2001) |
| Biogenic emission | MEGAN biogenic emissions online based upon the weather, |
| | land use data (Guenther et al., 2006) |
| Biomass burning | Biomass burning emissions and plume rise calculation |
| Dry deposition | Dry deposition of gas and aerosol species |
| Dust | GOCART dust emissions with AFWA modifications (Ginoux et al., 2001) |
| **Input data** | |
| Boundary cond. meteorology | ERA-Interim (Dee et al., 2011), resolution: 0.75°x 0.75°, |
| | 37 vertical levels from surface to 1 hPa |
| Sea surface temperature (SST) | NOAA OI SST (Reynolds et al., 2007) |
| Land use | USGS |
| Albedo | NCEP |
| Anthropogenic emissions | EDGAR HTAP (Janssens-Maenhout et al., 2015) |
| Boundary conditions chemistry | MOZART (Global CTM) |





**Table 2.** WRF and WRF-Chem simulations.

| Name | Description | Resolution | Period |
|---|---|---|---|
| **WRF_ref** | Nested WRF simulation | | |
| WRF_ref_D01 | (meteorology only) | Domain 01 (D01) 15x15 km$^2$ | 01-06/2013 |
| WRF_ref_D02 | model setup as in Tab. 1 | Domain 02 (D02) 3x3 km$^2$ | 01-06/2013 |
| | reference simulation | | |
| **WRFchem_ref** | Nested WRF-Chem simulation | | |
| WRFchem_ref_02_D01 | (including aerosol and chemistry) | Domain 01 (D01) 15x15 km$^2$ | 02/2013 |
| WRFchem_ref_02_D02 | model setup as in Tab. 1 using | Domain 02 (D02) 3x3 km$^2$ | 02/2013 |
| WRFchem_ref_05_D01 | EDGAR HTAP v2.2 emissions | Domain 01 (D01) 15x15 km$^2$ | 05/2013 |
| WRFchem_ref_05_D02 | | Domain 02 (D02) 3x3 km$^2$ | 05/2013 |
| **WRFchem_BC** | Nested WRF-Chem simulation | | |
| WRFchem_BC_02_D01 | (including aerosol and chemistry) | Domain 01 (D01) 15x15 km$^2$ | 02/2013 |
| WRFchem_BC_02_D02 | model setup as in Tab. 1 using | Domain 02 (D02) 3x3 km$^2$ | 02/2013 |
| WRFchem_BC_05_D01 | updated emission flux for black carbon | Domain 01 (D01) 15x15 km$^2$ | 05/2013 |
| WRFchem_BC_05_D02 | | Domain 02 (D02) 3x3 km$^2$ | 05/2013 |

**Table 3.** Overview and description of the measurement stations (T = temperature, WS = wind speed, WD = wind direction).

| Station number | Longitude [°] | Latitude [°] | Altitude [m] observations, D01, D02 | Source | Measured and analyzed parameters, availability of data in % based on hourly data |
|---|---|---|---|---|---|
| **1206** | 86.50 | 27.32 | 1720, 1558, 1571 | DHM | 2m T (100), 10m WS (100), 10m WD (100) |
| **1030** | 85.37 | 27.70 | 1337, 1407, 1315 | DHM | 10m WS (95) |
| **1015** | 85.20 | 27.68 | 1630, 1464, 1653 | DHM | 2m T (70), 10m WS (74), 10m WD (75) |
| **0909** | 84.98 | 27.17 | 130 , 159, 137 | DHM | 10m WS (84), 10m WD (84) |
| **0804** | 84.00 | 28.22 | 827 , 1053, 864 | DHM | 2m T (86) |
| **0017** | 85.38 | 27.68 | 1326, 1407, 1326 | SusKat | 2m T (71), 10m WS (91), 10m WD (91), RR (100), MLH (64) |
| **0014** | 85.31 | 27.72 | 1380, 1464, 1301 | SusKat | 2m T (77), 10m WS (78), 10m WD (77) |
| **42379** | 83.37 | 26.75 | | IGRA | T and relative humidity |
| **42182** | 77.2 | 28.58 | | IGRA | T and relative humidity |





**Table 4.** Black carbon emission fluxes per month used in the two simulations WRFchem_ref and WRFchem_BC for the area of the Kathmandu Valley.

| Month | EDGAR HTAP v2.2 $[\mathrm{ng\,m^{-2}\,s^{-1}}]$ | Estimated BC emission flux $[\mathrm{ng\,m^{-2}\,s^{-1}}]$ |
|---|---|---|
| **February 2013** | 28 | 196 |
| **May 2013** | 19 | 137 |

**Table 5.** Statistical overview of the model performance averaged over the time period January - June 2013 and all available stations based on 3-hourly data. Station measurements are included in the statistics if the data availability is over 70 % (Tab. 3).

| | Observations | WRF_ref_D02 | WRF_ref_D02 corrected | | Observations | WRF_ref_D02 |
|---|---|---|---|---|---|---|
| **Temperature** | | | | **Wind speed** | | |
| Mean [°C] | 17.8 | 18.6 | 18.5 | Mean [m s$^{-1}$] | 1.7 | 2.7 |
| Min/Max [°C] | 13.6 / 23.2 | 14.3 / 23.4 | 14.6 / 22.7 | Min/Max [m s$^{-1}$] | 0.6 / 3.5 | 0.8 / 5.5 |
| RMSE [°C] | - | 3.1 | 3.0 | RMSE [m s$^{-1}$] | - | 2.2 |
| Correlation | - | 0.9 | 0.9 | Correlation | - | 0.4 (0.1 - 0.6) |

**Table 6.** Number of observed and forecast precipitation days (days with sum of precipitation >0.5 $\mathrm{mm\,day^{-1}}$) during the period January - June 2013. Yes / yes - both data sets have a precipitation day at the same time; yes / no - first data set has a precipitation day, second does not; no / yes - first has no precipitation day, second has; no / no - both don't have a precipitation day. FAR - false-alarm ration, CSI - ciritical success index, H - hit ratio

| | yes/yes | yes/no | no/yes | no/no | FAR [%] | CSI [%] | H [%] |
|---|---|---|---|---|---|---|---|
| **Station measurement / TRMM** | 40 | 16 | 19 | 106 | 32 | 53 | 71 |
| **Station measurement / WRF_ref_D02** | 36 | 22 | 17 | 106 | 32 | 48 | 62 |
| **TRMM / WRF_ref_D02** | 34 | 26 | 19 | 102 | 36 | 43 | 57 |





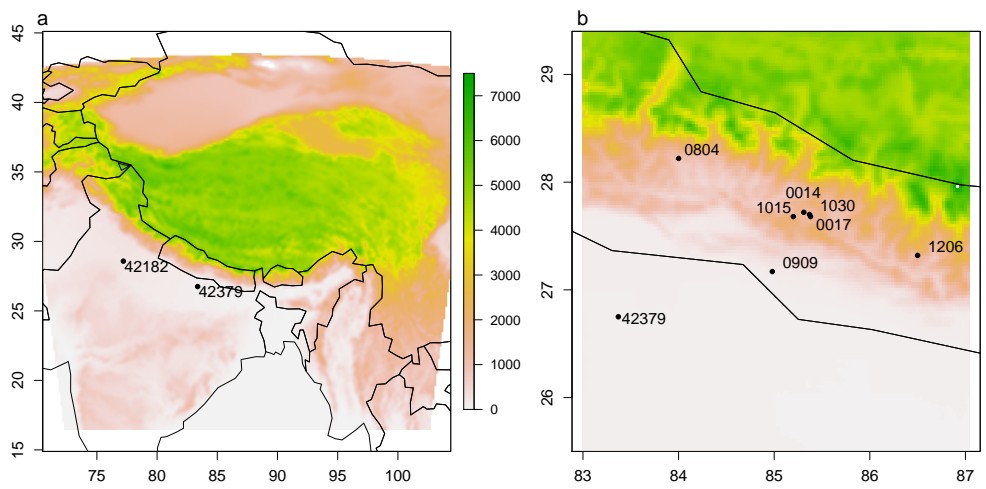

**Figure 1.** Model domains D01 (a) and D02 (b) as used in the WRF and WRF-Chem simulations. Shown are the terrain heights [m] and the locations and station numbers of the measurements sites.





**Figure 2.** Zonal and meridional wind fields in 500 hPa averaged over February and May 2013 for the WRF_ref_D01 simulation (a, c, e, g) and from the ERA-Interim reanalysis (b, d,f, h) in [$\mathrm{m\,s^{-1}}$].





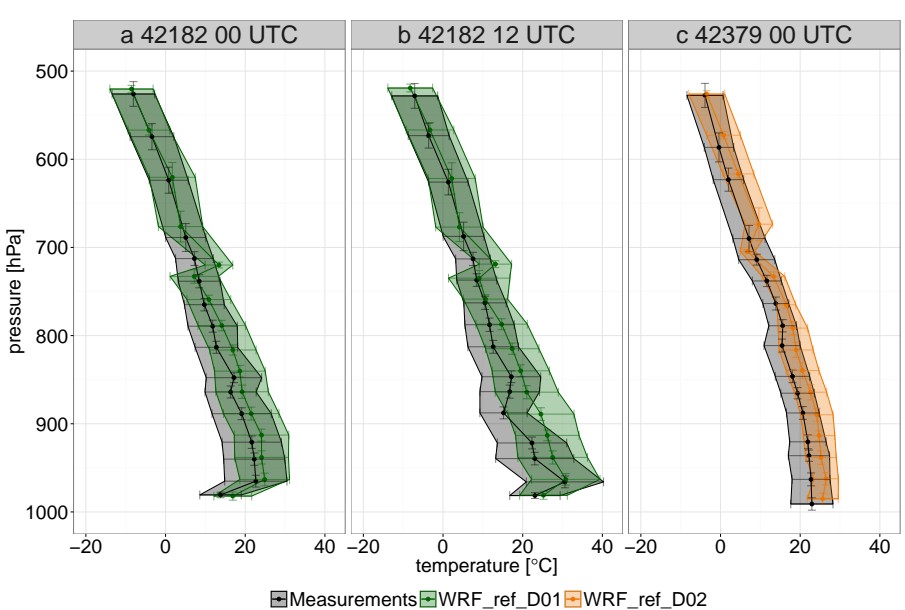

**Figure 3.** Averaged vertical profiles derived from radiosonde data and WRF simulations for temperature [°C] for the period January - June 2013. The figures show the results for the stations 42182 at 00 (a) and 12 UTC (b) and 42379 at 00 UTC (c). The shaded areas show the standard deviation, indicating the variability over the whole time period within each bin.





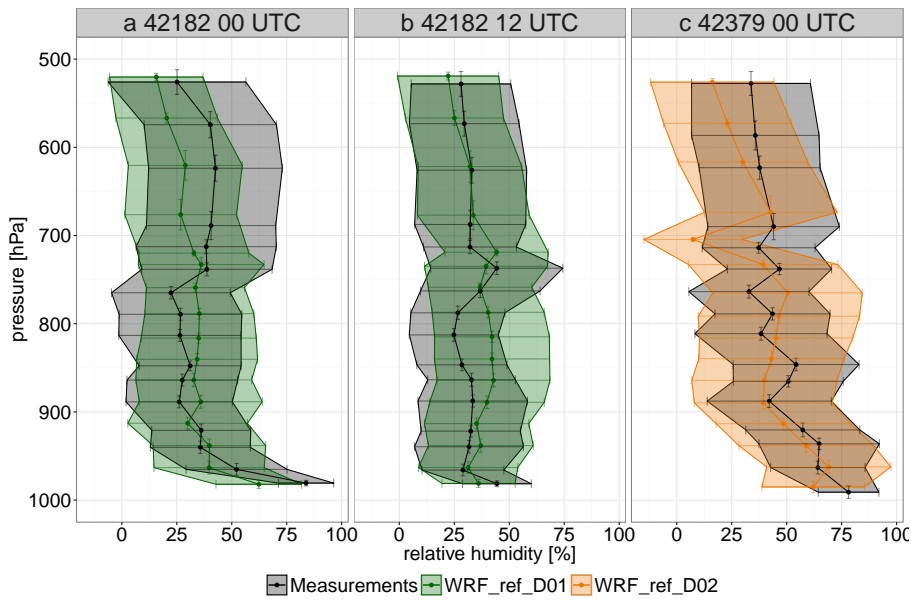

**Figure 4.** Averaged vertical profiles derived from radiosonde data and WRF simulations for relative humidity [%] for the period January - June 2013. The figures show the results for the stations 42182 at 00 (a) and 12 UTC (b) and 42379 at 00 UTC (c). The shaded areas show the standard deviation, indicating the variability over the whole time period within each bin.



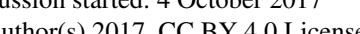

**Figure 5.** Time series of measured, simulated (WRF_ref_D02) and simulated but height corrected (WRF_ref_D02_corr) daily mean 2m temperature [°C] during January - June 2013 at the station 0804 (a), 1015 (b), 0014 (c), 0017 (d) and 6480 (e). The tables in the subfigures give the temporal correlation and the mean bias between simulated and measured values [°C].



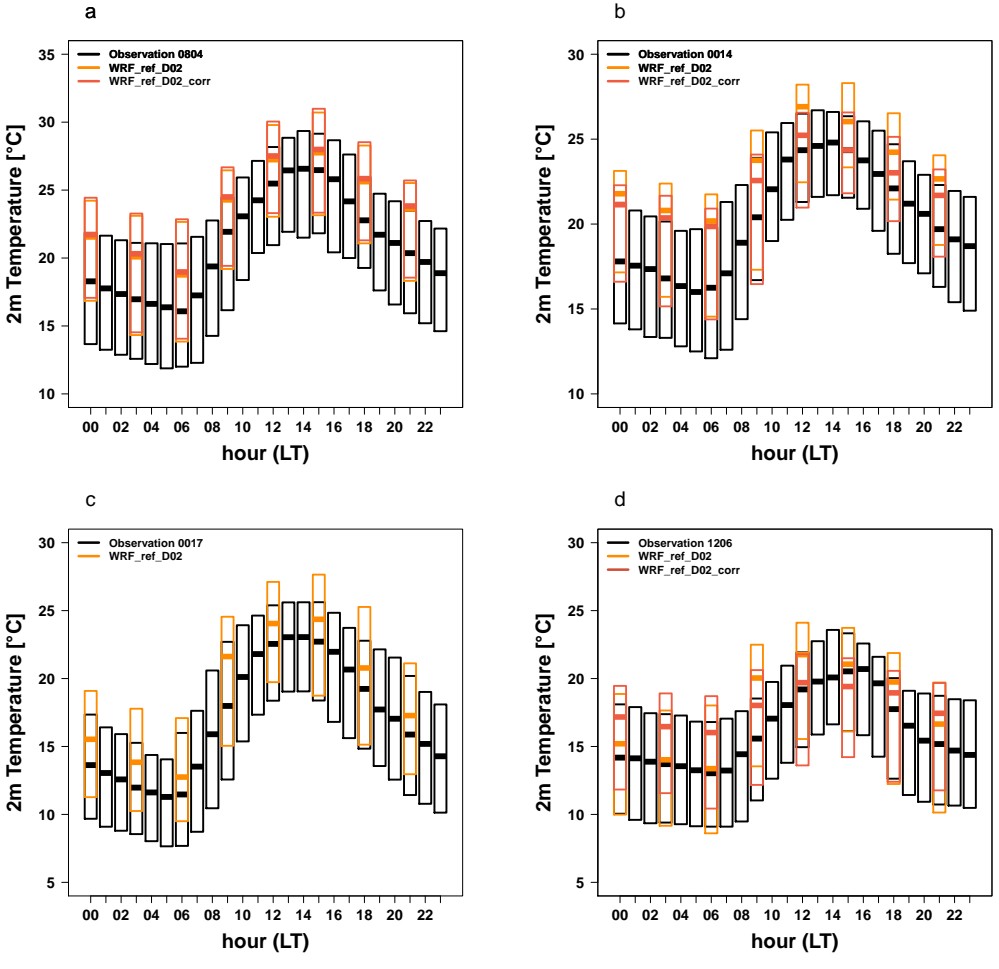

**Figure 6.** Diurnal cycle of the measured, simulated (WRF_ref_D02) and simulated but height corrected (WRF_ref_D02_corr) 2m temperature [°C] for the period January - June 2013 as a box-plot (showing the median, the upper and lower quantile) at the station 0804 (a), 0014 (b), 0017 (c) and 6480 (d).





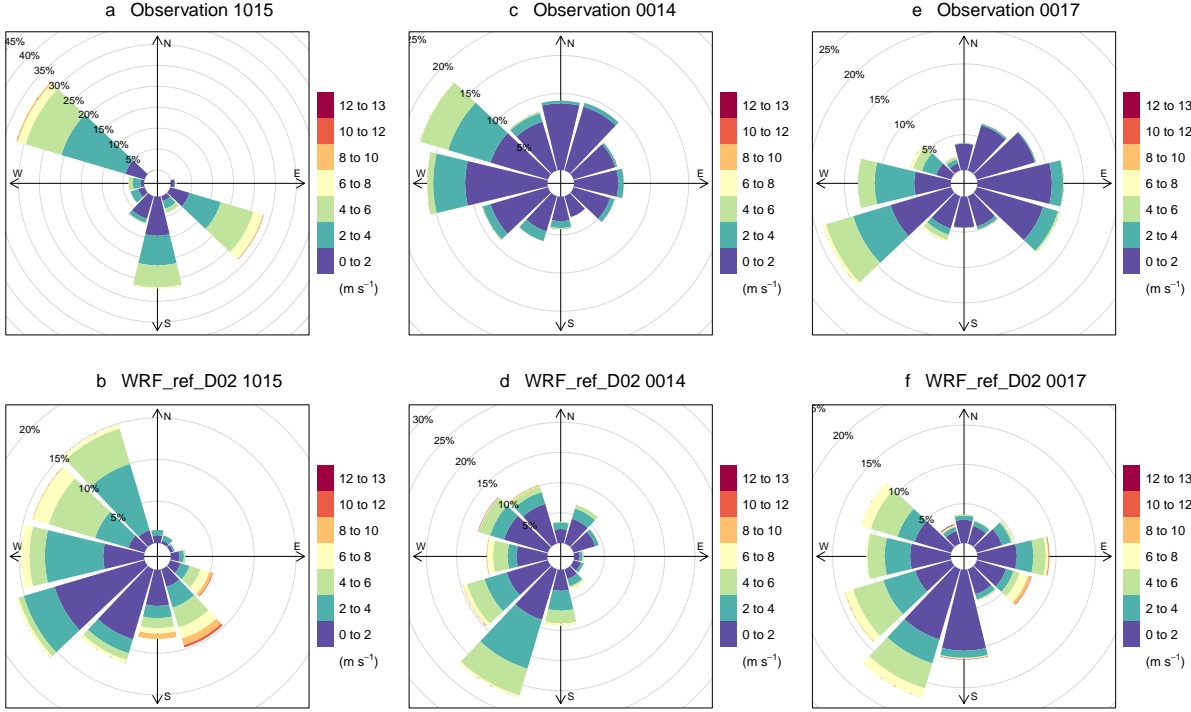

**Figure 7.** Wind roses based on measured and simulated (WRF_ref_D02) wind speed and direction at four stations (0018 (a, b), 1015 (c, d), 0014 (e, f) and 0017 (g, h)) in the Kathmandu Valley for the time period January - June 2013 based on 3-hourly data. Shown are wind speed (color) $[\mathrm{m\,s^{-1}}]$ and the frequncy of counts by wind direction [%].





a

b

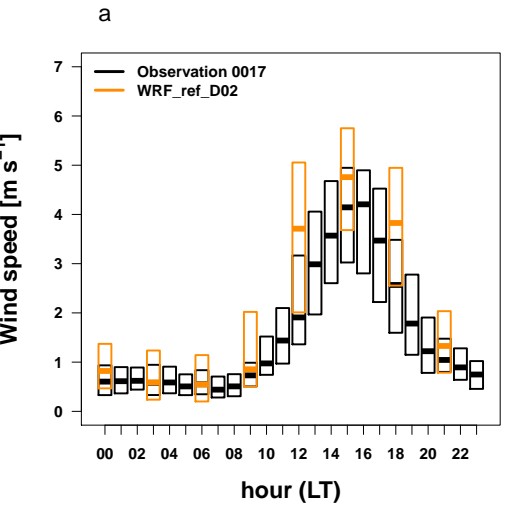
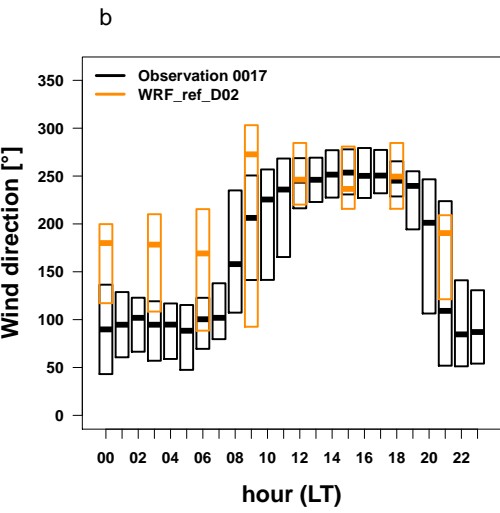

**Figure 8.** Diurnal cycle of the measured and simulated (WRF_ref_D02) wind speed [m s$^{-1}$] (a) and wind direction [°] (b) for the period January - June 2013 as a box-plot (showing the median, the upper and lower quantile) at the Bode station.





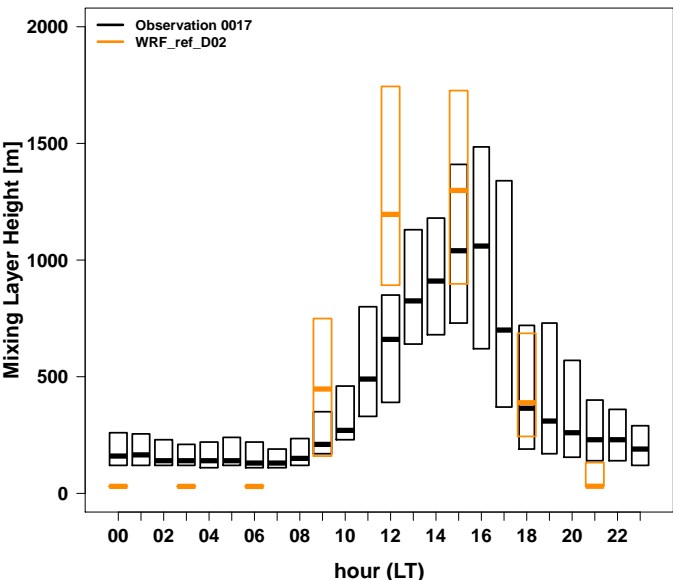

**Figure 9.** Diurnal cycle of the mixing layer heights [m] as a box-plot (showing the median, the upper and lower quantile) as diagnosed by the WRF model (WRF_ref_D02) and as determined from ceilometer measurement data at the Bode site for the period January - June 2013.





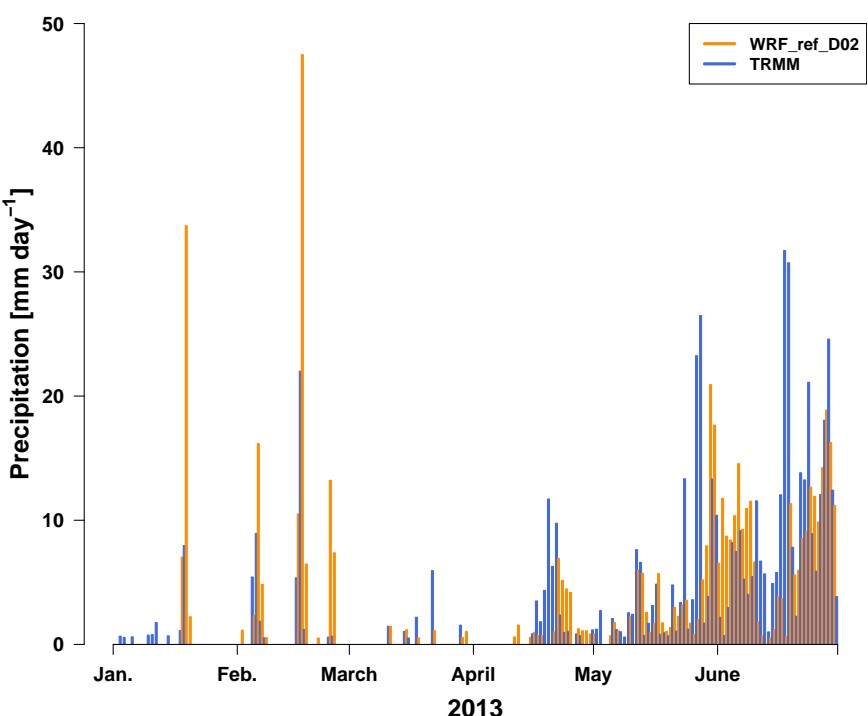

**Figure 10.** Timeseries of precipitation $[\mathrm{mm\,day^{-1}}]$ averaged over the domain D02 from WRF_ref_D02 and TRMM per day for January - June 2013 based on daily sums.



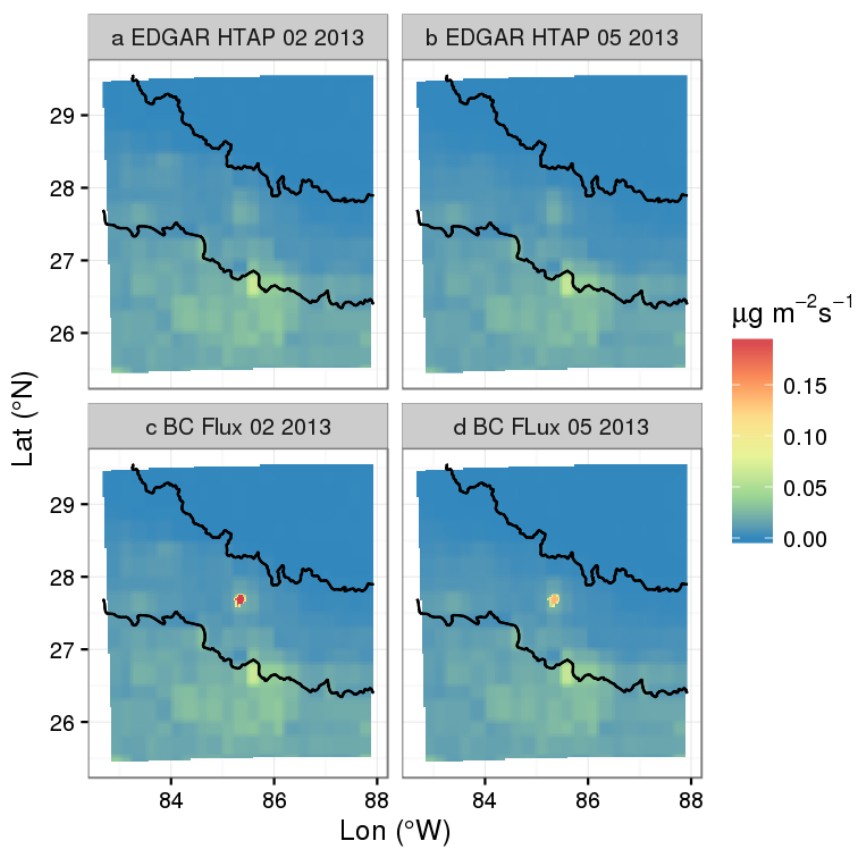

**Figure 11.** Black carbon emission flux used for the WRFchem_ref_02/05_D02 (a, b) and WRFchem_BC_02/05_D02 (c, d) simulations for February (left) and May 2013 (right) in $\mu g\,m^{-2}\,s^{-1}$.



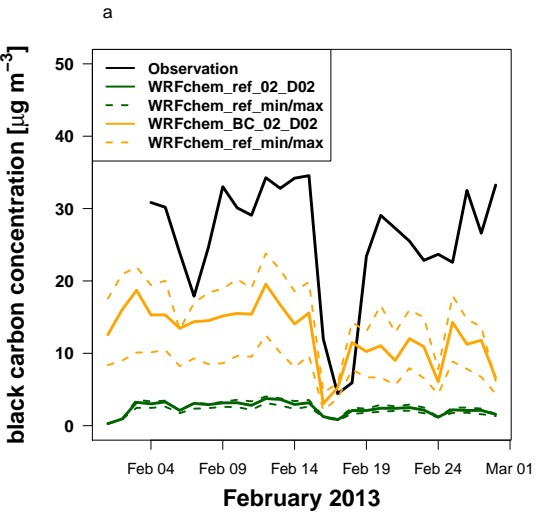
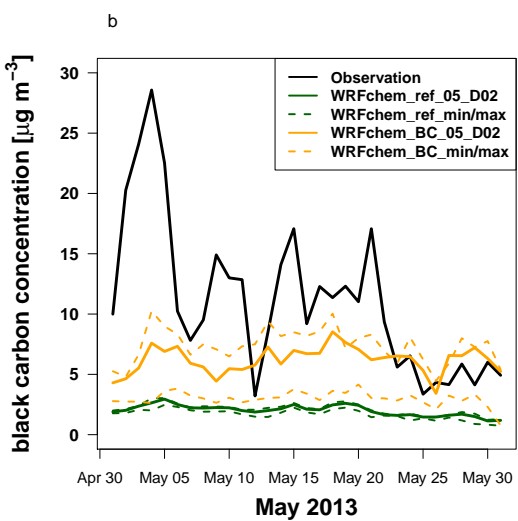

**Figure 12.** Time series of daily mean measured and simulated (WRFchem_ref_02/05_D02, WRFchem_BC_02/05_D02) black carbon concentrations [μg m$^{-3}$] at the Bode station for February (a) and May 2013 (b).

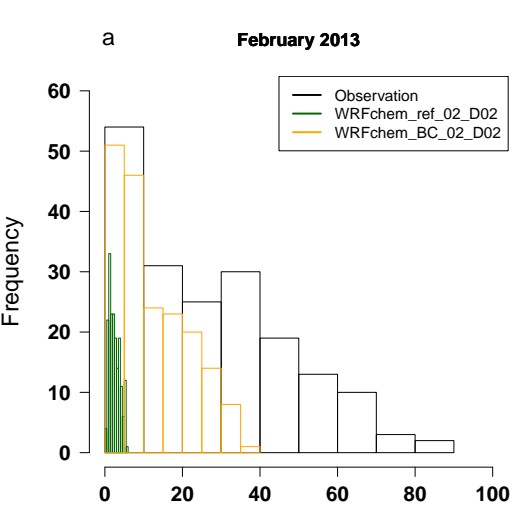

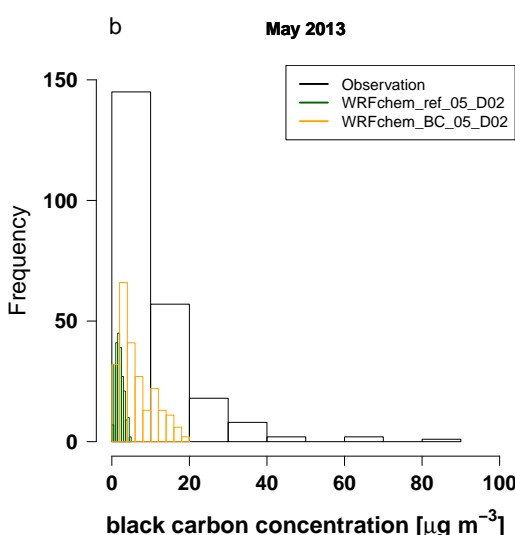

**Figure 13.** Black carbon concentrations at the Bode site, measured and simulated with WRF-Chem for February 2013 WR-Fchem_ref_02/05_D02 (a) and for May 2013 WRFchem_ref_02/05_D02 (b) as a histogram calculated from the 3-hourly values.





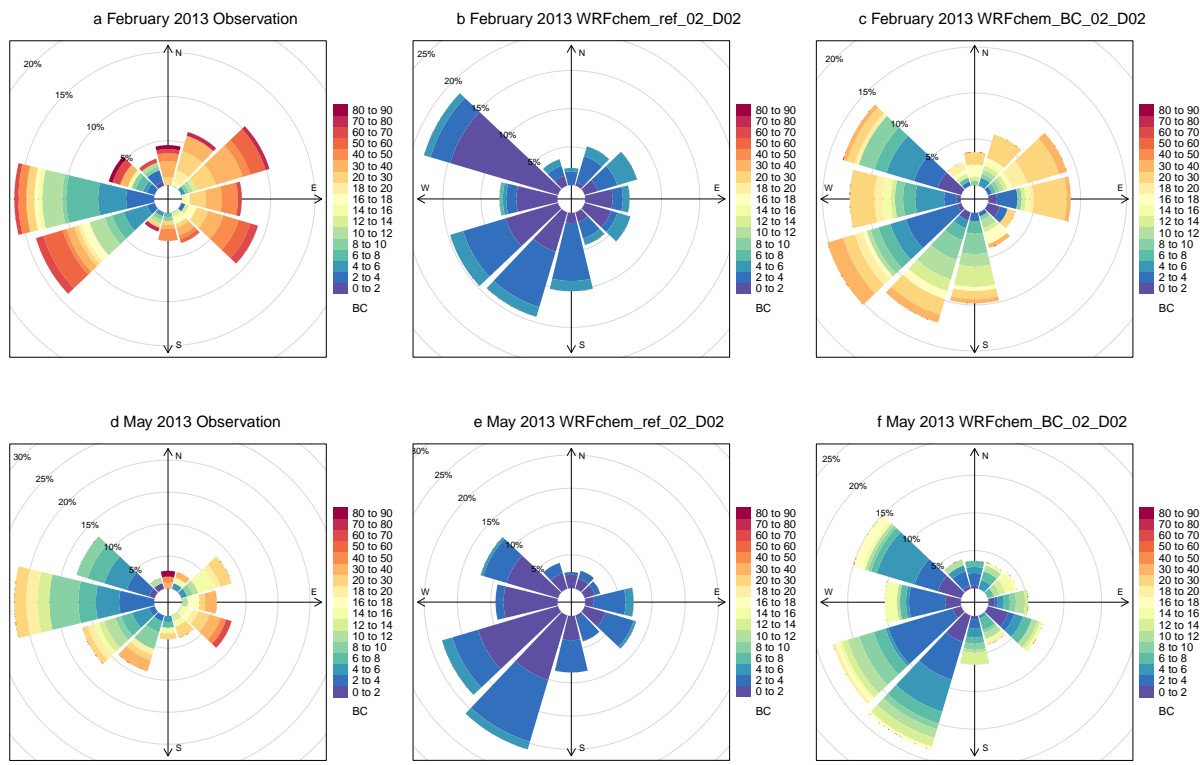

**Figure 14.** Pollution rose for black carbon at the Bode site calculated from the measured and simulated (WRFchem_ref_02/05_D02 and WRFchem_BC_02/05_D02) 3-hourly values of black carbon, wind speed and direction in February (a, b, c) and May (d, e, f) 2013. The figures represents the black carbon concentrations which coincide with a certain wind direction at the station and the frequency of occurrence of the wind direction in percent.