# Peer review of "WRF and WRF-Chem v3.5.1 simulations of meteorology and black carbon concentrations in the Kathmandu Valley"

_Geoscientific Model Development, 2017_

## Short Comment (SC1) · 5 Oct 2017

Dear authors,

in my role as Executive editor of GMD, I would like to ask you to include also the version number of WRF and WRF-Chem in the title of your manuscript upon submission of the revised version. Even if the model code version number is not yet mandatory for the evaluation paper type, the model version number is important information as evaluation results might change drastically between different versions of the same model.

Best regrads,

Astrid Kerkweg

---

## Referee Comment (RC1) · Anonymous Referee #1 · 13 Nov 2017

Mues et al. present evaluation of meteorological variables, and black carbon (BC) simulated by WRF-Chem model in the Kathmandu. Since the region is experiencing strong anthropogenic emissions, and that the surrounding topography is also highly complex, evaluation of model performance is valuable.

The paper is recommended for publication in GMD, however several comments and suggestions as listed below should be considered during the revision.

Title: Air quality is used in more general context, and it is expected to have study of more pollutants, while here focus is only BC. Possibly t would be better to revise the paper title as e.g. WRF-Chem simulations of meteorology and black carbon concentrations in kathmandu valley

Page 3, l.29: what is the motivation behind preventing sea salt emissions from the small in land lakes? What is the effect on results presented, if this change is not made in the model?

Page 6, l.16 and section 3.1.1: How does model winds compare with the reanalysis near surface (e.g. 800 hPa)? as compared to those currently presented (500 hPa)

Page 7, l.24-28: this should be part of data description (2.3.4), and not the evaluation metrics.

section 3.1.2. The text is too general. Please discuss by using average values and standard deviations at few representative pressure levels.

3.1.5. and Fig. 9: if there are sufficient observational data to show the comparison separately for winter (Jan-feb) and pre-monsoon(March-June), that would be more informative. Singh et al. (Atmos. Chem. Phys., 2016) using this model found overestimation of boundary layer especially towards the pre-monsoon in northern India /Himalayas. It should be mentioned if it is similar / or different at Kathmandu.

3.2.2 It would be possibly useful to discuss the uncertainties among different available emission inventories of BC in this region, and that whether the modified emission flux (based on observations) is within these uncertainties or not. Sharma et al. (Atmos. Chem. Phys. Discuss., 2017, in this special issue) showed large differences among different recent inventories of pollutants over South Asian regions.

Are there BC observations at other stations too in Kathmandu to evaluate further the simulation using observation-based emission flux?

Page 17, l.35, page 18: 1-2: Could authors show a comparison of model and observational BC concentrations separately for day and night, to indicate which sources could particularly be underestimated.

---

## Referee Comment (RC2) · Anonymous Referee #2 · 16 Nov 2017

This paper presents an evaluation of a modelling experiment of the meteorology and the surface concentrations of black carbon (BC) in a region of South Asia and Nepal, the Kathmandu Valley. The authors apply the WRFv3.5.1 meteorological model and the WRF-Chemv3.5.1 online meteorology-chemistry model over two domains centred over the Kathmandu Valley, the region of study. High-resolution simulations are conducted for the first half of 2013 year covering the same period of the experimental campaign SusKat-ABC. Emissions from EDGAR HTAP v2.2 database and an updated estimation over the Kathmandu Valley are used for the chemistry, and the meteorology is initialized with ERA-Interim meteorological reanalysis.

[Figure]

Although the objective of the study is of relevance for the scientific community (complexity of the region, high concentration of pollutants, lack of modelling efforts conducted in the area) the objectives, methodology and results presented in the manuscript do not fulfil the scope of the Geoscientific Model Development Journal. No model developments are presented nor discussed in the manuscript, although the authors claim that they have introduced relevant improvements to the WRF-Chem model, and the work is mainly an evaluation exercise of preliminary results with the WRF-Chem model perturbing global emissions available over the region. The experiments need a much in depth work in the emissions applied to run the full-chemistry of WRF-Chem. It is criticisable the use of a global inventory as EDGAR HTAP v2.2 to run a high-resolution mesoscale chemistry model as WRF-Chem at 3 km x 3 km horizontal resolution without significantly complementing the inventory with more detailed data (improving emission sectors and temporal profiles). From the results, it is clear that a significant lack in emission sources is the main limitation of the study, although an initial effort is done including the estimation of emissions of the Kathmandu Valley from Mues et al. (2017). Relevant emissions for BC that should be refined from HTAP data for India and Nepal domains are those associated with biomass burning, emissions from stoves, kerosene lamps, flaring gas or open burning of domestic waste. Current global aerosol models present large underestimations of AOD and BC surface concentrations in South Asia and the region of study, being most of the systems based on HTAP emissions (i.e., AEROCOM phase III experiment). A mesoscale chemistry model with such emissions won't be able to reproduce the huge concentrations of BC without a significant work in emission estimates. Without proper emissions the discussion of BC dispersion will be fault of information.

A part from the BC experiment, the first part of the manuscript is devoted to the evaluation of wind, temperature, relative humidity, mixing height and precipitation of the WRFv3.5.1 model applied over the region of study. The authors use a small set of observations available in the region, and discuss the results based on averages for the period of study. Again, there is no model development in this work and no clear recommendations for improving the meteorological model can be raised from the discussion. Although the analysis with monthly averages and daily means simplifies the description of results, no clear outcomes can be derived from the analysis.

In my opinion, this paper deserves a major revision before considering to be published in Geoscientific Model Development. The authors should consider introducing a more clear description of the model developments done, significantly improve the emissions used with WRF-Chem, and extend the meteorological analysis to present contributions that improve the meteorological model. I do not consider that specific comments are needed at this stage if the previous considerations are not addressed in a revised manuscript.

—————————————————————

---

## Referee Comment (RC3) · Anonymous Referee #3 · 27 Nov 2017

**Review of "Air quality in the Kathmandu Valley: WRF and WRF-Chem simulations of meteorology and black carbon concentrations" by Mues et al.**

**General comments**
This paper presents evaluation of the WRF simulated meteorology and WRF-Chem simulated black carbon mass concentrations in Kathmandu Valley. Air quality has been degrading rapidly in the Kathmandu Valley and thus needs immediate attention so that the public health can be protected from acute air pollution episodes. In this direction, this study takes a much-needed step and advances the community efforts by complementing intensive observations collected during the SusKat-ABC filed campaign with chemistry transport modeling. The Kathmandu Valley is a very difficult area to model mainly due to complex topography and wide range of emission sources active in the Valley. Therefore, modeling studies like this are essential to establish credibility of the models before they can be employed in design of pollution control strategies. The paper is very well written and easy to understand. However, I think this paper can benefit from a few WRF sensitivity experiments (see my first specific comment) and thus recommend major revision. My recommendation does not mean that the paper has scientific or technical flaws but it is to assure that the authors have sufficient time to perform and analyze the suggested WRF experiments.

**Specific comments**
I have third specific comments listed below.

First, I was surprised to see large differences between WRF and ERA-Interim wind fields (Figure 2) because WRF is driven by the ERA-Interim itself. Since the model runs are a month long, I think WRF is drifting away significantly from the large-scale forcing provided by the ERA-Interim. Thus, I suggest the authors to conduct a model experiment by nudging the WRF meteorological fields towards the ERA-Interim above the planetary boundary layer, and examine if that helps in reducing the bias. In case, the authors are not aware of the nudging option in WRF, here are the steps to run analysis nudging in WRF (http://www2.mmm.ucar.edu/wrf/users/wrfv2/How_to_run_grid_fdda.html). In addition to this, I think the authors also need to examine the sensitivity of model results to land use in WRF. The USGS land-use category used here is representative of 1994 and Kathmandu Valley has changed dramatically since then. Thus, I suggest conducting a WRF simulation with MODIS land-use. MODIS land-use is representative of 2003 but this experiment should still help us understand the sensitivity of model results to land-use representation.

Second, I think section 3.2.3 needs further detailed investigation. I believe that this event is potentially driven by open burning of agricultural crop residue in northern part of India and forest fires in Himalayas. The failure of the model to capture this event should not be attributed only to the anthropogenic emissions in Kathmandu Valley. It is important to understand the relative importance of local vs. non-local sources in this event as well as uncertainties in biomass-burning emissions. I realize that such an exercise can be time-consuming and can lead to another paper in itself. Thus, I recommend deleting this section. However, I suggest the authors to include a discussion about the potential impact of uncertainties in open biomass burning emissions and long-range transport on black carbon mass concentrations in the Kathmandu Valley.

Third, I recommend the authors to quantitatively assess the model performance by comparing their statistical metrics for temperature and wind speed against the benchmarks by Emery (2001). This is important for this paper as the focus is on evaluating the meteorological parameters that are highly relevant to air quality.

**Minor Comments:**
Page 1, Line 20: Change "long-term" to "extensive" because 6 months is not long-term.

Page 3, Line 8: I think it is important to state how different regional and global models have performed in simulating BC mass concentrations in South Asia. This will nicely connect the present study to literature. Here are few studies that employed regional and global models to simulate black carbon mass concentrations in South Asia [e.g., Ganguly et al., 2009; Nair et al., 2012; Moorthy et al., 2013; Pan et al., 2015; Kumar et al., 2015a, 2015b, Goverdhan et al., 2016]

Page 9, Line 28: This is probably a typo here because there is no panel corresponding to station "1206" in Figure 5.

Section 3.1.6: I suggest adding a map of the WRF and TRMM precipitation for February and May so that readers can visualize if the model is able to simulate the precipitation in right places.

Table 1: Please name the inventory used to represent biomass burning emissions.

Figure 12: Should the last legend read as "WRFchem_BC_min/max"?

**References**

Emery, C. A.: Enhanced meteorological modeling and performance evaluation for two Texas ozone episodes, Prepared for the Texas Natural Resource Conservation Commission, by ENVIRON International Corporation, 2001.

Ganguly, D., et al.: Inferring the composition and concentration of aerosols by combining AERONET and MPLNET data: Comparison with other measurements and utilization to evaluate GCM output. J. Geophys. Res. 114, D16203 (2009), doi:10.1029/2009JD011895.

Govardhan, G., Nanjundiah, R. S., Satheesh, S. K., Moorthy, K. K., and Kotamarthi, V. R.: Performance of WRF-Chem over Indian region: Comparison with measurements. J. Earth Sys. Sci. 124 (4), 875-896 (2015), doi: 10.1007/s12040-015-0576-7.

Kumar, R., et al.: Sources of black carbon aerosols in South Asia and surrounding regions during the Integrated Campaign for Aerosols, Gases and Radiation Budget (ICARB). Atmos. Chem. Phys. 15, 5415-5428 (2015a), doi:10.5194/acp-15-5415-2015.

Kumar, R., et al.: What controls the seasonal cycle of black carbon aerosols in India?, J. Geophys. Res. Atmos. 120, 7788–7812 (2015b), doi:10.1002/2015JD023298.

Nair, V. S., Solmon, F., Giorgi, F., Mariotti, L., Babu, S. S., and Moorthy, K., K.: Simulation of South Asian aerosols for regional climate studies. J. Geophys. Res. 117, D04209 (2012), doi:10.1029/2011JD016711.

Pan, X., et al.: A multi-model evaluation of aerosols over South Asia: common problems and possible causes. Atmos. Chem. Phys. 15, 5903-5928 (2015), doi:10.5194/acp-15-5903-2015.

Moorthy, K. K., et al.: Performance evaluation of chemistry transport models over India. Atmos. Environ. 71, 210–225 (2013), doi: 10.1016/j.atmosenv.2013.01.056.

---

## Author Comment (AC1) · 27 Nov 2017

We thank reviewer #2 for reading our manuscript. We also thank the reviewer for sharing their concerns about the appropriateness of our manuscript, but strongly disagree with the reviewer's point of view that our article is not within the scope of Geoscientific Model Development because it *"is mainly an evaluation exercise"* and *"no model developments are presented"*. One of the six manuscript types listed on the GMD website is "model evaluation papers". Our manuscript has been submitted as such, as can be seen by the "MS Type". Furthermore, we would like to stress that our manuscript is a contribution to the special issue "The community version of the Weather Research and

[Figure]

Forecasting Model as it is coupled with Chemistry (WRF-Chem)". This special issue "[...] hosts scientific technical documentation and evaluation manuscripts concerned with the community version of WRF-Chem."

We also disagree with the reviewer's opinion that "[...] *the authors claim that they have introduced relevant improvements to the WRF-Chem model* [...]". In the manuscript we state that (page 3, line 27) "Two modifications have been applied to WRF-Chem compared to the standard model version." regarding emission of sea salt and gravitational settling of aerosol particles. Both modifications are not relevant to simulations of black carbon over Nepal and have been mentioned in the manuscript for completeness. No claims of significant model improvements have been made.

In this study, the WRF-Chem model is used to examine to which extent a widely used state-of-the-art meteorology and air quality model is able to reproduce observations in the Kathmandu region, and provide a preliminary diagnosis (not a full scale investigation) of where there are still gaps in in our understanding of emissions and processes. We clearly highlight that there is a significant gap in the emissions, and addressing that gap is beyond scope of the paper. A comprehensive emission inventory for Nepal including previously under-characterized sources is currently being developed and will be used in future publications to understand the role of emissions (or the influences of different emission inventories). We therefore also disagree with the reviewer's opinion that *"The authors should* [...] *significantly improve the emissions used with WRF-Chem, and extend the meteorological analysis to present contributions that improve the meteorological model."* as this is not the focus and also not within the scope of a model evaluation study, rather the study provides information to the community about where model deficiencies are found and what developments (e.g., emissions datasets) need to be prioritized.

---

## Editor Comment (EC1) · S. Marras (Editor) · 30 Nov 2017

Dear authors,

The open discussion has been closed.

Two of the three reviewers recommended major revisions and requested to review the revised version of the paper. To promptly proceed with the evaluation of your manuscript and its potential publication, please address all of the reviewers' comments and requests.

Best regards. Simone Marras

---

## Author Comment (AC2) · 13 Mar 2018

Below we address the comment of the executive editor raised during the open discussion of the paper "Air quality in the Kathmandu Valley: WRF and WRF-Chem simulations of meteorology and black carbon concentrations". We have listed the editor's comment below and our answer is provided in blue.

Executive Editor

[Figure]

Dear authors,
in my role as Executive editor of GMD, I would like to ask you to include also the version number of WRF and WRF-Chem in the title of your manuscript upon submission of the revised version. Even if the model code version number is not yet mandatory for the evaluation paper type, the model version number is important information as evaluation results might change drastically between different versions of the same model.
Best regrads,
Astrid Kerkweg

As requested, we added the model version to the title now reading "WRF and WRF-Chem v3.5.1 simulations of meteorology and black carbon concentrations in the Kathmandu Valley"

---

## Author Comment (AC3) · 13 Mar 2018

Below we address the comments of reviewer #1 and questions raised during the open discussion of the paper "Air quality in the Kathmandu Valley: WRF and WRF-Chem simulations of meteorology and black carbon concentrations". We would like to thank the reviewer for the time and effort reviewing the paper. We feel it has improved thanks to the constructive comments. We have listed all reviewer comments below and our answers are provided in blue. A "track changes" version of the revised manuscript is provided as a supplement with all changes to the manuscript highlighted.

Anonymous Referee #1

The paper is recommended for publication in GMD, however several comments and suggestions as listed below should be considered during the revision. Title: Air quality is used in more general context, and it is expected to have study of more pollutants, while here focus is only BC. Possibly t would be better to revise the paper title as e.g. WRF-Chem simulations of meteorology and black carbon concentrations in kathmandu valley

We agree with the reviewer's point and changed the title into "WRF and WRF-Chem v3.5.1 simulations of meteorology and black carbon concentrations in the Kathmandu Valley" (see also comment of the executive editor).

Page 3, l.29: what is the motivation behind preventing sea salt emissions from the small in land lakes? What is the effect on results presented, if this change is not made in the model?

In the simulations shown in this paper, we use the sea salt emission scheme recommended in the WRF-Chem v3.5.1 user's guide for the selected chemical mechanism RADM2/SORGAM (seas_opt = 2). This sea salt scheme does not distinguish between ocean and freshwater grid cells (lakes) resulting in unrealistic emissions from lakes in the Himalayas when running the model at high horizontal resolution (at low resolution, the inland lakes are not resolved). The correction applied here is more of cosmetic nature as sea salt aerosol does not play an important role in the Kathmandu Valley.

Page 6, l.16 and section 3.1.1: How does model winds compare with the reanalysis near surface (e.g. 800 hPa)? as compared to those currently presented (500 hPa)

[Figure]

We added discussion of the 800 hPa winds to section 3.1.1 (zonal and meridional wind fields) and added the corresponding figure to the supplementary material (figure S1).

Page 7, l.24-28: this should be part of data description (2.3.4), and not the evaluation metrics.

As suggested, we moved the corresponding paragraph from section 2.4 (evaluation metrics) to section 2.3.4 (radiosonde data).

section 3.1.2. The text is too general. Please discuss by using average values and standard deviations at few representative pressure levels.

We extended the discussion of the vertical profiles in section 3.1.2 focusing on some selected pressure levels.

3.1.5. and Fig. 9: if there are sufficient observational data to show the comparison separately for winter (Jan-feb) and pre-monsoon (March-June), that would be more informative. Singh et al. (Atmos. Chem. Phys., 2016) using this model found overestimation of boundary layer especially towards the pre-monsoon in northern India /Himalayas. It should be mentioned if it is similar / or different at Kathmandu.

Following the suggestion of the reviewer, we extended section 3.1.5 (mixing layer height) and now discuss winter (January-February) and the pre-monsoon season (March-June) separately. A comparison with the results from Singh et al. (2016) has also been included and figure 9 has been updated accordingly.

3.2.2 It would be possibly useful to discuss the uncertainties among different available emission inventories of BC in this region, and that whether the modified emission flux (based on observations) is within these uncertainties or not. Sharma et al. (Atmos. Chem. Phys. Discuss., 2017, in this special issue) showed large differences among different recent inventories of pollutants over South Asian regions.

The observation-based emission estimate ranges between 64 and 248 ng m$^2$ s$^{-1}$ depending on the time of the year. In contrast, the BC emission flux in the Kathmandu Valley in the EDGAR HTAP emission database, which is based on the REAS data in this region, ranges between 19 and 28 ng m$^2$ s$^{-1}$. The INTEX-B emission dataset gives 21 ng m$^2$ s$^{-1}$ for the Kathmandu Valley. Such a comparison is already discussed in Mues et al. (2017) and we think a repetition of this discussion is not needed here. Instead, we added a reference to this discussion in Mues et al. (2017) as well as a reference to the work of Jayarathne et al. (2018) to section 3.2.2 (discussion of the observation-based emission estimates for black carbon). We think the measurements of numerous emission factors in Nepal by Jayarathne et al. (2018) is an important contribution to reduce the large uncertainties in future emission inventories for this region.

Are there BC observations at other stations too in Kathmandu to evaluate further the simulation using observation-based emission flux?

Besides Bode, BC concentrations were also measured at Pakanajol, a site near the center of the Kathmandu Metropolitan City about 9km (aerial distance) to the northwest of the Bode site. The BC concentrations at both sites were found comparable in all seasons (Putero et al., Aerosol Air qual. Res., 2015; Putero et al., Atmos. Chem.

Phys., 2018) and have therefore not been used in this study. We added this information to the revided manuscript, section 2.3.1 (SusKat-ABC field campaign).

Page 17, l.35, page 18: 1-2: Could authors show a comparison of model and observational BC concentrations separately for day and night, to indicate which sources could particularly be underestimated.

We extended the corresponding paragraph in section 4 (summary and outlook) by adding a discussion on the difference between night-time and all-day biases in the modeled BC concentration.

---

## Author Comment (AC4) · 13 Mar 2018

Addition to our reply to the comments of reviewer #2

Following the reviewer's comment to extend the meteorological analysis, we included/extended the evaluation of meteorological parameters by adding:

- geographical distribution of precipitation (section 3.1.6) and a new figure (fig. 11) showing a comparison of precipitation maps from WRF and TRMM

- mixed layer height (section 3.1.5) by adding a separate discussion of the dry and rainy season and by revising figure 9 now showing the diurnal cycle of MLH for these two seasons separately

- vertical profiles of temperature and humidity from radiosondes (section 3.1.2)

- zonal and meridional wind fields (section 3.1.1) by adding an analysis of the 800 hPa wind fields (new figure S1 in the supplement)

- comparison of the statistical metrics used for temperature and wind speed with the benchmarks proposed by Emery et al. (2001) (sections 3.1.3 and 3.1.4)

A "track changes" version of the revised manuscript is provided as a supplement. Please also see our 'Reply to reviewer #2' published on 27 Nov 2017 in the GMD discussion forum.

---

## Author Comment (AC5) · 13 Mar 2018

Below we reply to the anonymous referee #3's comments and questions on our GMDD manuscript "Air quality in the Kathmandu Valley: WRF and WRF-Chem simulations of meteorology and black carbon concentrations". We would like to thank the reviewer for the constructive comments helping us to improve the paper. We have listed all reviewer comments below and answers are provided in blue. A "track changes" version of the revised manuscript is provided as a supplement with all changes to the manuscript highlighted.

Anonymous Referee #3

Specific comments

I have third specific comments listed below.

First, I was surprised to see large differences between WRF and ERA-Interim wind fields (Figure 2) because WRF is driven by the ERA-Interim itself. Since the model runs are a month long, I think WRF is drifting away significantly from the large-scale forcing provided by the ERA-Interim. Thus, I suggest the authors to conduct a model experiment by nudging the WRF meteorological fields towards the ERA-Interim above the planetary boundary layer, and examine if that helps in reducing the bias. In case, the authors are not aware of the nudging option in WRF, here are the steps to run analysis nudging in WRF (http://www2.mmm.ucar.edu/wrf/users/wrfv2/How_to_run_grid_fdda.html). In addition to this, I think the authors also need to examine the sensitivity of model results to land use in WRF. The USGS land-use category used here is representative of 1994 and Kathmandu Valley has changed dramatically since then. Thus, I suggest conducting a WRF simulation with MODIS land-use. MODIS land-use is representative of 2003 but this experiment should still help us understand the sensitivity of model results to land-use representation.

Following the reviewer's suggestions, we performed two additional sensitivity simulations, one in which the meteorological fields were nudged above the boundary layer, and one that involved the use of MODIS land use data. We have summarized our findings in a new subsection:

"3.1.7 Sensitivities of main meteorological parameters to nudging technique and land use data

In order to test that the simulated large-scale circulation does not drift or deviate from the observed synoptic condition, a sensitivity simulation in which a grid nudging technique was employed for horizontal winds, temperature and water vapor above boundary layer has been performed. In this simulation, we obtained similar results as in the reference simulation, for example the RMSE of temperature is 3.0 K using the nudging approach compared to 3.1 K in the reference run. The model performance for wind speed does not change. In the upper troposphere the differences in the simulated meteorological variables in the reference and the sensitivity runs were statistically insignificant, suggesting that the WRF model results in this altitude range are mostly driven by the prescribed boundary conditions. In a second sensitivity simulation we have analyzed the impact using of MODIS land use data instead of the default USGS dataset. In this simulation the impact of using the MODIS data together with applying the nudging technique on WRF results is tested for temperature and wind speed parameters. As in the first sensitivity simulation, the RMSE of temperature does not deviate much from the one obtained from the reference simulation, i.e. using USGS land use data and no nudging, leading to a RMSE of 2.9 K compared with 3 K in the WRF_ref_D02 simulation. In contrast to temperature, the model performance for wind speed worsens with a RMSE of 3.2 m s$^{-1}$ and an average correlation coefficient of 0.21. Since the relative small number of measurement stations in the evaluation domain might not be representative for the whole domain, we have also compared the results from the sensitivity simulations with the reference simulation. When applying the nudging technique the domain averaged mean bias between the sensitivity and the reference simulation -0.03 K for temperature and 0.08 m s$^{-1}$ for wind speed. For the MODIS land use sensitivity simulation the domain averaged mean bias when compared to the reference simulation is 0.08 K for temperature and 0.2 m s$^{-1}$ for wind speed. This suggests that the changes in temperature and wind speed when applying

the nudging technique and using the MODIS land use dataset are rather small and not expected to be important factors in explaining the differences between the model results and observations found."

Second, I think section 3.2.3 needs further detailed investigation. I believe that this event is potentially driven by open burning of agricultural crop residue in northern part of India and forest fires in Himalayas. The failure of the model to capture this event should not be attributed only to the anthropogenic emissions in Kathmandu Valley. It is important to understand the relative importance of local vs. non-local sources in this event as well as uncertainties in biomass-burning emissions. I realize that such an exercise can be time-consuming and can lead to another paper in itself. Thus, I recommend deleting this section. However, I suggest the authors to include a discussion about the potential impact of uncertainties in open biomass burning emissions and long-range transport on black carbon mass concentrations in the Kathmandu Valley.

Following the reviewer's suggestion, we deleted section 3.2.3 (Case study: the episodes 2-5 May and 6-8 May 2013).
The biomass burning emissions in the model are calculated from satellite observations of fires and land cover using average emission factors (Fire Inventory from the National Center for Atmospheric Research (NCAR) version 1: FINN, Wiedinmyer et al., 2011). We think that anything beyond the rather general uncertainty analysis given in Wiedinmyer et al. (2006 and 2011) would not only be beyond the scope of this study but also beyond our expertise.

Third, I recommend the authors to quantitatively assess the model performance by comparing their statistical metrics for temperature and wind speed against the benchmarks by Emery (2001). This is important for this paper as the focus is on
evaluating the meteorological parameters that are highly relevant to air quality.

Following the recommendation of the reviewer, we have included the benchmark proposed by Emery et al. (2001) in the discussions of the model performance in sections 2.4 (evaluation metrics), 3.1.3 (2m temperature) and 3.1.4 (10m wind speed and direction).

Minor Comments:

1. Page 1, Line 20: Change "long-term" to "extensive" because 6 months is not long-term.

Changed as suggested.

2. Page 3, Line 8: I think it is important to state how different regional and global models have performed in simulating BC mass concentrations in South Asia. This will nicely connect the present study to literature. Here are few studies that employed regional and global models to simulate black carbon mass concentrations in South Asia [e.g., Ganguly et al., 2009; Nair et al., 2012; Moorthy et al., 2013; Pan et al., 2015; Kumar et al., 2015a, 2015b, Goverdhan et al., 2016]

As suggested, we added a paragraph to the introduction linking this work to studies with different regional and global models including references to Goverdhan et al. (2016), Nair et al. (2012), Pan et al. (2015) and Moorthy et al. (2013).

3. Page 9, Line 28: This is probably a typo here because there is no panel corresponding to station "1206" in Figure 5.

Thanks for spotting this. The typo was actually not in the text but in the caption of figure 5 (and figure 6). We corrected "6480" to "1206" in both cases.

4. Section 3.1.6: I suggest adding a map of the WRF and TRMM precipitation for February and May so that readers can visualize if the model is able to simulate the precipitation in right places.

As suggested, we added a figure comparing the precipitation from WRF and TRMM in February and May (new figure 11). The discussion in section 3.1.6 (precipitation) has been extended with a brief discussion of the new figure.

5. Table 1: Please name the inventory used to represent biomass burning emissions.

For biomass burning emissions, the Fire Inventory from the National Center for Atmospheric Research (NCAR) version 1 (FINN, Wiedinmyer et al., 2011) is used. This has been added to table 1.

6. Figure 12: Should the last legend read as "WRFchem_BC_min/max"?

The legend has been corrected in the revised version of figure 12.